# Metabolic network percolation quantifies biosynthetic capabilities across the human oral microbiome

David B Bernstein[1,2], Floyd E Dewhirst[3,4], Daniel Segrè[1,2,5,6,7]*

[1]Department of Biomedical Engineering, Boston University, Boston, United States; [2]Biological Design Center, Boston University, Boston, United States; [3]The Forsyth Institute, Cambridge, United States; [4]Harvard School of Dental Medicine, Boston, United States; [5]Bioinformatics Program, Boston University, Boston, United States; [6]Department of Biology, Boston University, Boston, United States; [7]Department of Physics, Boston University, Boston, United States

**Abstract** The biosynthetic capabilities of microbes underlie their growth and interactions, playing a prominent role in microbial community structure. For large, diverse microbial communities, prediction of these capabilities is limited by uncertainty about metabolic functions and environmental conditions. To address this challenge, we propose a probabilistic method, inspired by percolation theory, to computationally quantify how robustly a genome-derived metabolic network produces a given set of metabolites under an ensemble of variable environments. We used this method to compile an atlas of predicted biosynthetic capabilities for 97 metabolites across 456 human oral microbes. This atlas captures taxonomically-related trends in biomass composition, and makes it possible to estimate inter-microbial metabolic distances that correlate with microbial co-occurrences. We also found a distinct cluster of fastidious/uncultivated taxa, including several *Saccharibacteria* (TM7) species, characterized by their abundant metabolic deficiencies. By embracing uncertainty, our approach can be broadly applied to understanding metabolic interactions in complex microbial ecosystems.
DOI: https://doi.org/10.7554/eLife.39733.001

*For correspondence:
dsegre@bu.edu

**Competing interests:** The authors declare that no competing interests exist.

## Introduction

Metabolism, in addition to enabling growth and homeostasis for individual microbes, contributes to the organization of complex, dynamic microbial communities. Within these communities, different microbes have diverse metabolic capabilities that lead to interactions driving microbial community structure and dynamics at multiple spatial and temporal scales (*Ponomarova and Patil, 2015*; *Phelan et al., 2012*; *Watrous et al., 2013*; *Harcombe et al., 2014*; *Embree et al., 2015*). For example, through cross-feeding, a compound produced by one species might benefit another, leading to a network of metabolic interdependences (*Embree et al., 2015*; *Goldford et al., 2017*; *Mee et al., 2014*; *Pande et al., 2015*; *D'Souza et al., 2018*; *Zengler and Zaramela, 2018*; *Pacheco et al., 2019*; *Mee and Wang, 2012*). This type of interaction has been proposed as one of the main reasons for the prevalence, in natural microbial communities, of uncultivated (or fastidious) microbes (*Stewart, 2012*; *Epstein, 2013*; *Pande and Kost, 2017*; *Staley and Konopka, 1985*). These microbes do not grow in pure culture on standard laboratory conditions as they may depend on diffusible metabolites produced by neighboring microbes (*Pande and Kost, 2017*). The prominence of uncultivated/fastidious microbial organisms across the tree of life and their potential importance in microbial community structure is highlighted by the recent identification of the candidate phyla radiation – a large branch of the tree of life consisting mainly of uncultivated organisms with small

genomes and unique metabolic properties (*Kantor et al., 2013*; *Brown et al., 2015*; *Hug et al., 2016*). Efforts towards understanding this important component of microbial communities require further knowledge of metabolic interdependencies driven by biosynthetic deficiencies.

Some of the most promising strides in understanding metabolic interdependences between microbes have been taken in the study of the human oral microbiome. The human oral microbiome serves as an excellent model system for microbial communities research, due to its importance for human health and ease of access for researchers (*Dewhirst et al., 2010*; *Wade, 2013*). For example, the order of colonization of species in dental plaque has been characterized physically (*Kolenbrander et al., 2010*) and metabolically (*Mazumdar et al., 2013*), and visualized microscopically (*Mark Welch et al., 2016*). The human oral microbiome consists of roughly 700 different microbial species, identified by 16S rRNA microbiome sequencing and cataloged in the human oral microbiome database (*Dewhirst et al., 2010*; *Chen et al., 2010*). Importantly, 63% of species in the human oral microbiome have been sequenced, including several uncultivated and recently-cultivated strains implicated in oral health and disease (*Krishnan et al., 2017*; *Siqueira Jr and Rôças, 2013*). Exciting recent work has led to successful laboratory co-cultivation of at least three previously uncultivated organisms, the *Saccharibacteria* (TM7) phylum taxa: *Saccharibacteria* bacterium HMT-952 strain TM7x (*Bedree et al., 2018*; *He et al., 2015*; *Bor et al., 2016*; *Bor et al., 2018*), *Saccharibacteria* bacterium HMT-488 strain AC001 (*Collins et al., 2019a*), and *Saccharibacteria* bacterium HMT-955 strain PM004 (*Collins et al., 2019b*). *Saccharibacteria* are prominent in the oral cavity and relevant for periodontal disease (*Brinig et al., 2003*; *Ouverney et al., 2003*). Due to their importance, they were among the first uncultivated organisms from the oral microbiome to be fully sequenced via single-cell sequencing methods (*Marcy et al., 2007*), and represent the first co-cultivated members of the candidate phyla radiation (*He et al., 2015*). Thus, their metabolic and phenotypic properties are of great interest for oral health and microbiology in general.

In parallel to achieving laboratory growth of diverse and uncultivated bacteria, a major unresolved challenge is understanding the detailed metabolic mechanisms that may underlie their dependencies. Ideally, one would want to computationally predict, directly from the genome of an organism, its biosynthetic capabilities and deficiencies, so as to translate sequence information into mechanisms and community-level phenotypes (*Widder et al., 2016*). A number of approaches, based on computational analyses of metabolic networks, have contributed significant progress towards this goal (*Schuster et al., 2000*; *Oberhardt et al., 2009*; *Lewis et al., 2012*). At the heart of these methods are metabolic network reconstructions, formal encodings of the stoichiometry of all metabolic reactions in an organism, that are readily amenable to multiple types of in silico analyses and simulations (*Feist et al., 2009*). Recent exciting progress has led to the automated generation of 'draft' metabolic network reconstructions for any organism with a sequenced genome (*Henry et al., 2010*), opening the door for the quantitative study of large and diverse microbial communities. The most commonly used metabolic network analysis methods – flux balance analysis (FBA) (*Orth et al., 2010a*) and its dynamic version (dFBA) (*Mahadevan et al., 2002*) – have been extensively applied to study microbial communities (*Harcombe et al., 2014*; *Embree et al., 2015*; *Pacheco et al., 2019*; *Magnúsdóttir et al., 2017*; *Magnúsdóttir and Thiele, 2018*; *Zarecki et al., 2014*; *Stolyar et al., 2007*; *Klitgord and Segrè, 2010*; *Freilich et al., 2011*; *Zelezniak et al., 2015*; *Cook and Nielsen, 2017*; *Biggs et al., 2015*; *Zomorrodi and Segrè, 2016*). However, FBA and dFBA are not easily applicable to automatically-generated draft metabolic networks due to gaps (missing or incorrect reactions) in the metabolic network, and are thus difficult to scale to large and diverse microbial communities. Methods for 'gap-filling' draft reconstructions can address this problem, and ensemble methods potentially present a promising approach (*Biggs and Papin, 2017*; *Machado et al., 2018*). However, any gap-filling comes at the expense of an increased risk for false positive predictions. Additionally, gap-filling typically requires specific knowledge or assumptions on the growth media composition – which are often difficult to obtain for diverse environmental isolates and by definition unknown for uncultivated organisms. Alternatively, topology-based metabolic network analysis methods, such as network expansion (*Ebenhöh et al., 2004*) and NetSeed-based methods (*Borenstein et al., 2008*), are less dependent on gap-filling and have been applied to the analysis of draft metabolic reconstructions. These methods have provided valuable large-scale insight into metabolic processes in microbial communities, including the biosynthetic potentials of organisms and metabolites (*Basler et al., 2008*; *Matthäus et al., 2008*), the chance of cooperation or competition between species (*Carr and Borenstein, 2012*; *Kreimer et al., 2012*; *Levy et al., 2015*;

*Opatovsky et al., 2018*), and the relationship between organisms and environment (*Borenstein et al., 2008*; *Freilich et al., 2009*; *Handorf et al., 2008*), for example in the human gut microbiome (*Levy and Borenstein, 2013*). While all of these approaches are promising, an additional issue that continues to limit the use of metabolic network analysis for prediction of biosynthetic capabilities is the difficulty of generating these predictions when the chemical environment of the microbes is unknown. In complex microbial communities, such as the human microbiome, the exact chemical composition of the environment is difficult to estimate, due both to the molecular complexity of the environment itself, and to the likely prevalence of secretions, lysing and cross-feeding within the community. Thus, the capacity to provide metabolic predictions based on unelaborated genome annotation, and on limited knowledge about an organism's growth environment remains an important open challenge.

Here we introduce a new method, which begins to address the above limitations, and provides a novel prediction of an organism's biosynthetic capabilities. Our method applies a probabilistic approach to define and compute a metric that estimates which metabolites, such as biomass components, are robustly synthesized by a given metabolic network and which would likely need to be supplied from the environment/community. Discrepancies in these calculated estimates between organisms can be used to generate hypotheses regarding microbial auxotrophy and metabolic exchange in microbial communities. Importantly, our metric has the capacity to estimate biosynthetic capabilities in spite of uncertainty about environmental conditions by randomly sampling many different possible nutrient combinations. In this study, we first demonstrated our method on *E. coli* to clarify its performance and interpretation. Next, we applied our method to a large number of organisms from the human oral microbiome, and predicted broad trends in biosynthetic capabilities associated with taxonomy and microbial co-occurrence. We further focused our analysis on uncultivated microorganisms, including three recently co-cultivated *Saccharibacteria* (TM7) strains. In addition to highlighting their biosynthetic deficiencies, we developed specific hypotheses for their metabolic exchange with growth-supporting partner microbes.

## Analysis method

Our newly developed method quantifies the robustness with which a given metabolic network can produce a given metabolite from variable metabolic precursors. In essence, we quantify a metabolic network specific metric for metabolite producibility by probabilistically sampling sets of possible environments. While the probabilistic sampling can be adjusted to reflect a specific environment, its power lies largely in the capacity to explicitly incorporate in a statistical way the lack of knowledge about environmental composition.

The inspiration for this method comes from the statistical physics concept of percolation. Percolation theory has been applied in a wide range of fields, including the study of cascading metabolic failure upon gene deletions in metabolism (*Smart et al., 2008*; *Barabási, 2015*). In percolation theory the robustness of a network can be characterized by randomly adding or removing components (nodes or edges) of a network and assessing network connectivity (*Barabási, 2015*). The smaller the number of components that need to be randomly added to the network before it becomes connected, the more robust it is to perturbations. We utilized this concept to characterize the network robustness of a particular metabolic network towards producing a specified target metabolite by randomly adding input metabolites to the network and assessing the network's ability to produce the target.

To implement our method, we first introduced a probabilistic framework for analyzing metabolic networks (*Figure 1* and *Figure 1—figure supplement 1*). In this framework, every metabolite can be considered to be drawn from a Bernoulli distribution, i.e. present in the network with a given input probability ($P_{in}$). These probabilities could represent beliefs about the environment, chances of metabolites being available from a host organism, or any arbitrary prior assumption on metabolite inputs. Throughout the majority of our analyses we have assigned $P_{in}$ to be an identical value for all input metabolites. However, as illustrated in an example in our results section (*Metabolite producibility in a protein vs. carbohydrate-enriched environment*) this probabilistic framework can utilize $P_{in}$ values that vary across metabolites. Following the assignment of $P_{in}$, the network structure is used to calculate the output probability ($P_{out}$) of some specified target metabolite. In practice, random sampling of probabilistically drawn input metabolite sets is used to calculate the probability of producing the target metabolite. For each random sample, a modified version of FBA (*Orth et al., 2010a*)

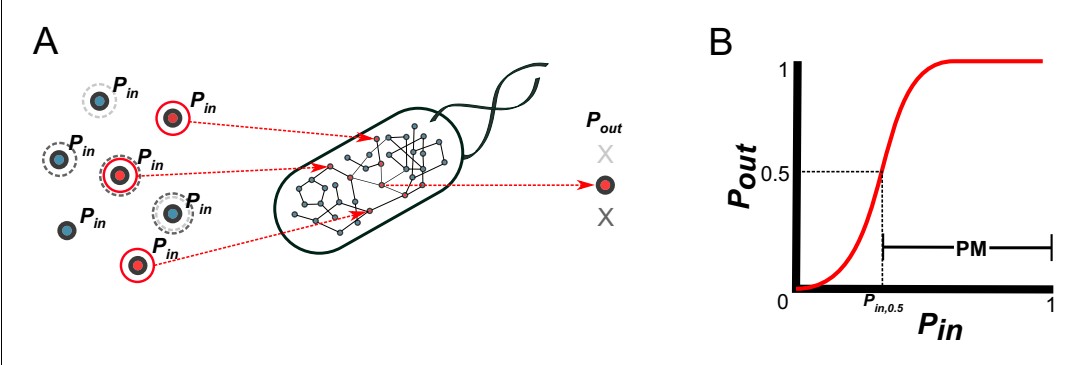

**Figure 1.** A probabilistic framework for calculating the producibility metric (PM). (A) Random samples of input metabolites are added to the metabolic network with probability $P_{in}$. Samples are shown here with gray or red circles. Sampled input metabolites are then used to calculate if a specified target output metabolite can be produced or not. Here the solid red circled sample leads to production of the target metabolite while the dotted gray circled samples do not. The probability of producing the target output metabolite ($P_{out}$) is calculated by taking many random samples at a specified $P_{in}$. (B) A producibility curve is calculated which represents $P_{out}$ as function of $P_{in}$. Points along this curve are sampled by assigning the $P_{in}$ value and estimating $P_{out}$. The $P_{in}$ value at which $P_{out} = 0.5$ ($P_{in,0.5}$) is used to define the producibility metric (PM) as PM = 1-$P_{in,0.5}$.

DOI: https://doi.org/10.7554/eLife.39733.002

The following figure supplements are available for figure 1:

**Figure supplement 1.** Probabilistic framework simple example.

DOI: https://doi.org/10.7554/eLife.39733.003

**Figure supplement 2.** Theoretical properties of the producibility curve.

DOI: https://doi.org/10.7554/eLife.39733.004

is used to assess the network's ability to produce the target metabolite (for a complete explanation of how FBA is implemented in this context, see methods section: Algorithm functions, *feas*).

Using the above probabilistic framework, we defined a novel metric quantifying biosynthetic capabilities, the producibility metric (PM) (*Figure 1B*). The PM is calculated as follows: First, a producibility curve describing $P_{out}$ as a function of $P_{in}$ is generated for a given metabolic network and metabolite target. This curve can be estimated by sampling input metabolites for different values of $P_{in}$ (between 0 and 1), and calculating $P_{out}$. Next, we calculated the $P_{in}$ value along the producibility curve at which $P_{out}$ is equal to 0.5 ($P_{in,0.5}$, analogous to the $K_m$ in the Michaelis-Menten curve). Finally, PM is defined as PM = 1-$P_{in,0.5}$, such that larger PM values correspond to increased robustness. Our method calculates PM efficiently by random sampling and a nonlinear fitting algorithm (for details, see methods section: Algorithm functions *calc_PM_fit_nonlin*). In addition to calculating PM computationally for arbitrary metabolic networks and metabolites, we also derived a way to calculate PM analytically using combinatorial equations. The combinatorial equations are built up from simple scenarios to the most general in *Figure 1—figure supplement 2*. This analytical result, verified in detail for one specific pathway (*Figure 2—figure supplement 2*) clarifies the connection between our metric and the concept of minimal precursor sets (*Andrade et al., 2016*). It describes mathematically how the PM captures the multiplicity of routes through which a given target metabolite can be produced, and could serve as the basis for further theoretical work on the fundamental properties of metabolic networks.

The algorithms used to implement our method are written in MATLAB and designed as a set of modular functions that interface with the COBRA toolbox – a popular metabolic modeling software compendium (*Schellenberger et al., 2011*; *Heirendt et al., 2019*). The methodology behind each function is further explained in the methods section. The code is freely available online at https://github.com/segrelab/biosynthetic_network_robustness (*Bernstein, 2019*; copy archived at https://github.com/elifesciences-publications/biosynthetic_network_robustness).

# Results

## Using the *E. coli* core metabolic network to demonstrate features of metabolite producibility

Before applying our approach to the systematic study of genome-scale metabolic networks from the human oral microbiome, we used the model organism *E. coli* to illustrate the performance and interpretation of our method. We started with the *E. coli* core metabolic network, a simplified network consisting of central carbon metabolism and lacking peripheral metabolic pathways, such as amino acid or cofactor biosynthesis (*Orth et al., 2010b*). We calculated the PM for all intracellular metabolites in this network using a uniform ensemble of environments (as described in the methods). The results are shown in *Figure 2A*, overlaid on the *E. coli* core metabolic network itself, with each node's color indicating its PM value and node size indicating its degree of connectivity. Consistent with the high connectivity of the *E. coli* core metabolic network, most metabolites have high PM values (PM >0.950). For example, the metabolites H$^+$ and pyruvate are both highly connected in the metabolic network and have high PM (PM = 0.968 and 0.952 respectively). However, the network

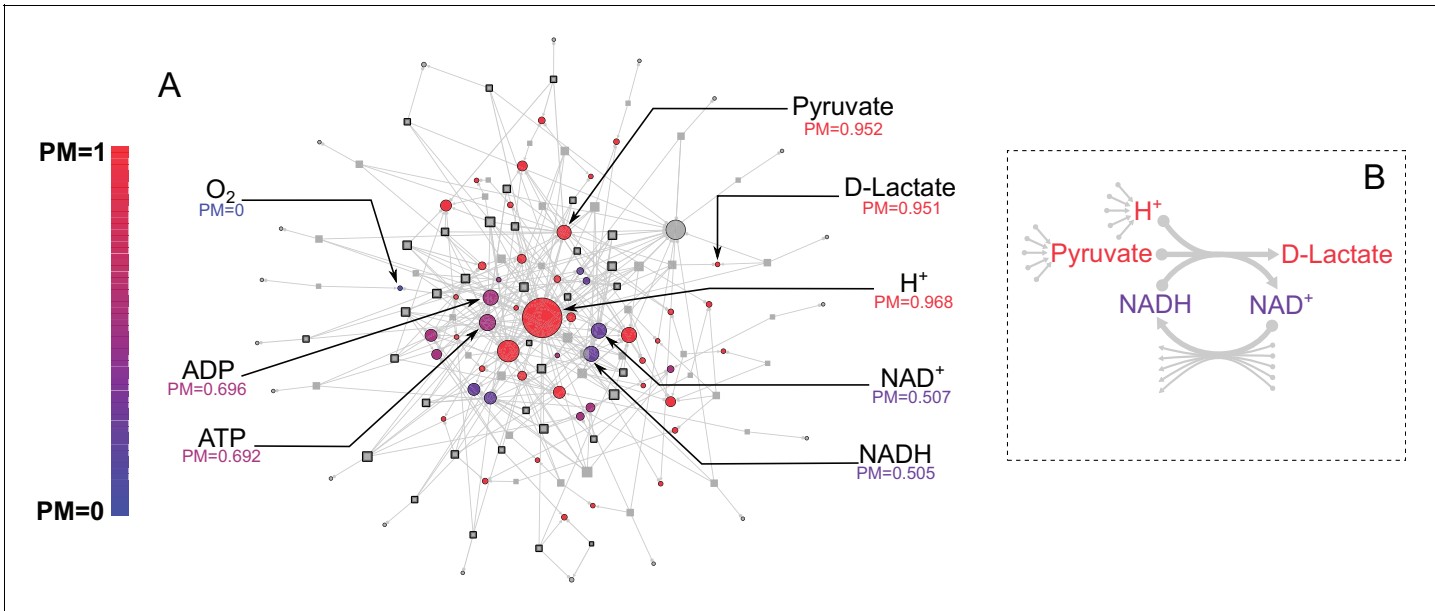

**Figure 2.** *E. coli* core metabolic network metabolite producibility. (**A**) The *E. coli* core metabolic network is represented as a bipartite graph with metabolites shown as circles and reactions shown as squares. Reactions shown with a black border are irreversible in the model, those with no border are reversible. All intracellular metabolites are colored based on their PM value (low – blue, high – red). Reactions and metabolite nodes are sized based on their total node degree. Several key metabolites of interest are highlighted with their corresponding PM values shown. Central metabolites such as H$^+$ and Pyruvate have high degree and high PM. Cofactors such as AMP/ADP/ATP and NAD$^+$/NADH have high degree but low PM, as they cannot be synthesized in this network. Oxygen is an example of a PM=0 metabolite that cannot be produced from any other metabolites in this network. D-lactate is an example of a metabolite with low degree and high PM that is it is easily produced but not well-connected. (**B**) The lactate dehydrogenase reaction producing D-Lactate is shown as an example to illustrate that poorly connected metabolites can display a high PM, and how recycled cofactors have minimal impact on PM values. Lactate dehydrogenase produces D-lactate and NAD$^+$ from pyruvate, H$^+$ and NADH. The metabolite D-lactate has high PM despite being produced only by this one reaction in the metabolic network because it can be produced from the high PM metabolites pyruvate and H$^+$, which are themselves produced from a large number of possible precursors. Although NADH is also used to produce D-lactate, and has a relatively low PM in this core model, it has minimal impact on the PM of D-lactate as NADH can be recycled from NAD$^+$ by a large number of reactions (represented by the arrows at the bottom of the figure) and thus production of NADH is not necessary for the production of D-lactate.

DOI: https://doi.org/10.7554/eLife.39733.005

The following figure supplements are available for figure 2:

**Figure supplement 1.** Node degree and producibility metric do not correlate for *E. coli* core metabolic network intracellular metabolites.
DOI: https://doi.org/10.7554/eLife.39733.006

**Figure supplement 2.** Producibility analysis of the histidine biosynthetic pathway.
DOI: https://doi.org/10.7554/eLife.39733.007

also contains several metabolites that are well connected, but have lower PM values. These include, for example, the cofactors AMP/ADP/ATP and NAD$^+$/NADH, which have PM values of ~0.7 and ~0.5 respectively, because they can be produced from each other, but not biosynthesized in this network. The network also includes several examples of metabolites that are poorly connected but have high PM values. One example is D-lactate, which is produced only via Lactate Dehydrogenase from the high PM metabolites Pyruvate and H$^+$ (*Figure 2B*). This reaction also consumes NADH and produces NAD$^+$ but because these cofactors can be easily recycled from each other by a large number of different reactions, their relatively low PM (as described above) has minimal influence on the PM value of D-lactate (*Figure 2B*). This example demonstrates the fact that our metric captures metabolites which are easily produced because their precursors are easily produced, and that the PM of recycled cofactors has minimal influence on the PM of a target metabolite. Overall, there is also no significant correlation between the PM values and the node degree of a metabolite in the network (*Figure 2— figure supplement 1*), indicating that our metric describes a more complex property of a metabolite in a network that is not captured simply by node degree.

## Producibility of metabolites differs from pathway completeness and captures minimal precursor set structure

We next applied our method in detail to a specific biosynthetic pathway within a genome-scale model to demonstrate how our PM provides information that is richer than what can be learned from simply counting the percent of reactions present in a given biosynthetic pathway. Specifically, we analyzed the histidine biosynthetic pathway in the curated *E. coli* iJO1366 genome-scale metabolic network (*Orth et al., 2011*), and checked how the two methods differ in their capacity to capture the effect of reaction knock-outs along the pathway (*Figure 2—figure supplement 2*). The PM is more sensitive than pathway completeness, as it captures features beyond the percent of reactions in the biosynthetic pathway. For different knockouts in the histidine biosynthetic pathway (which counts nine distinct reactions) the PM is related to the distance of the removed reaction from the target metabolite (histidine), whereas the completeness score would be the same (8 out of 9) for each auxotroph (*Figure 2—figure supplement 2B*). This same capacity of PM to capture finer details of the effect of missing reactions in a pathway is also confirmed by a similar analysis of histidine biosynthesis across all oral microbiome draft metabolic networks (oral microbiome network reconstruction and analysis described later in the manuscript) (*Figure 2—figure supplement 2C*).

In general, in contrast with the percent completion of the biosynthetic pathway, the PM depends deeply on the pathway structure, ultimately capturing the number of different routes through which the target metabolite can be synthesized (also called the minimal precursor sets; *Andrade et al., 2016*). This property stems intuitively from the way the PM is defined, and is explained precisely by our combinatorial theory (*Figure 1—figure supplement 2*). While our method's computational estimate of the PM is based on sampling the space of possible precursor sets, the combinatorial theory provides an exact value for the producibility of a molecule with a given minimal precursor set structure. The close match between the sample-based PM and the combinatorial theory for the histidine biosynthetic pathway (*Figure 2—figure supplement 2B*) suggests that the PM indeed captures the complex multiplicity of avenues for producing a given metabolite.

## Producibility analysis shows improved tolerance to missing reactions compared to flux balance analysis

One of the challenges we wished to address with our method is the possibility of making robust inferences about the metabolic capabilities of different organisms in spite of missing reactions – a situation often encountered upon reconstructing metabolic networks from newly sequenced genomes. To assess the performance of our approach in this context, we compared it with flux balance analysis (FBA) calculations for a genome-scale metabolic networks with a given number of randomly removed reactions. In particular, we applied both FBA and our method to the *E. coli* iJO1366 genome-scale metabolic network, which we gradually perturbed by removing an increasing number of randomly chosen reactions. In this performance test, the unperturbed iJO1366 metabolic network was used as a ground truth against which the predictions of our method and FBA on perturbed metabolic networks were compared. *Figure 3* shows the accuracy of both FBA and the PM as a function of the percentage of reactions removed from the iJO1366 metabolic network. While the output of

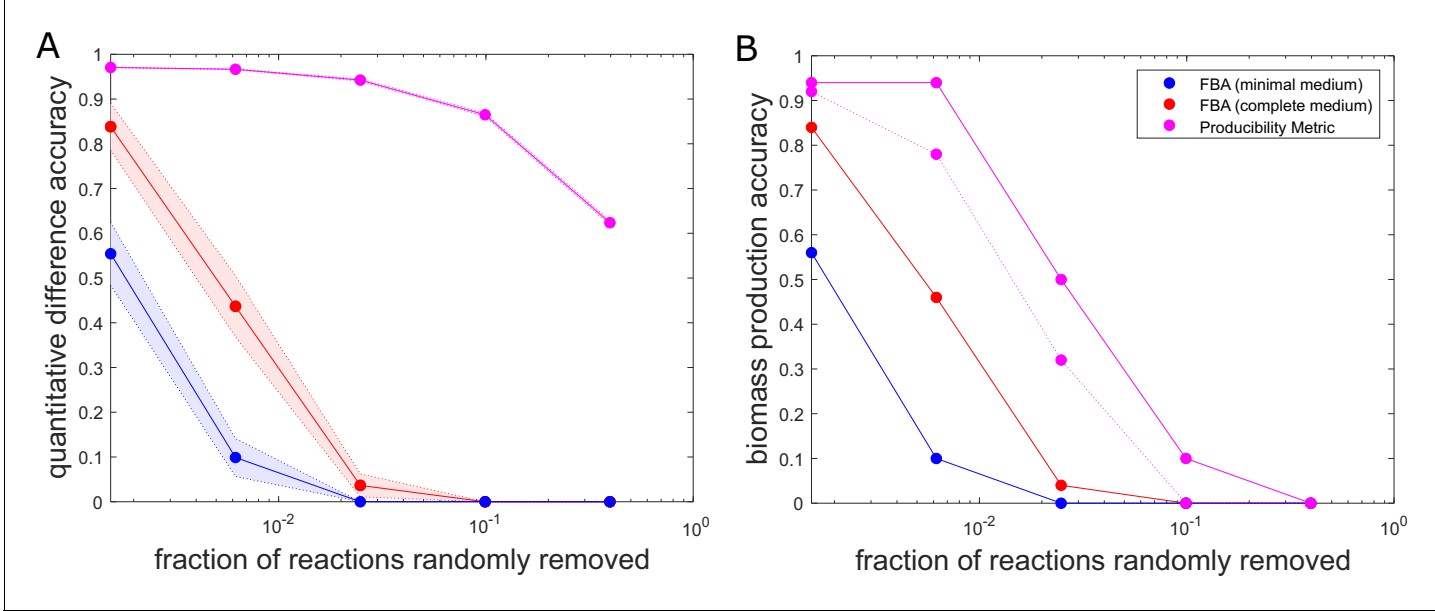

**Figure 3.** The accuracy of flux balance analysis and the producibility metric for different perturbed *E. coli* genome-scale metabolic networks. Reactions were randomly removed from the *E. coli* iJO1366 metabolic network generating 50 different networks at five different levels of reaction removal. These networks were then analyzed with the producibility metric (PM) and flux balance analysis (FBA) in a minimal and complete medium. The accuracy of the PM and FBA results were assessed through two different measures and plotted as a function of the number of reactions removed on a semi-log plot. (**A**) Quantitative difference accuracy – The accuracy was measured quantitatively based on the L1 norm of the difference between the original network metric and the randomly perturbed network metric. For FBA the L1 norm was computed as the absolute value of the difference between the biomass flux of the original network and the perturbed network. For the PM the L1 norm was calculated as the sum of the absolute value of the difference between each PM value. The L1 norm for both metrics was then normalized and subtracted from one to give a measure of accuracy. The mean of 50 different randomly perturbed networks at five different reaction removal levels is shown with dots connected by solid lines (FBA on minimal medium: Blue, FBA on complete medium: Red, PM: Purple). The standard error of the metric is shown as a shaded region around the line. (**B**) Biomass production accuracy – The accuracy was measured by the fraction of randomly perturbed metabolic networks that were capable of producing biomass. For FBA this was calculated as the fraction of networks capable of producing biomass flux above 1% of the unperturbed biomass flux (FBA on minimal medium: Blue, FBA on complete medium: Red). For the PM, the biomass production accuracy was calculated as the fraction of networks capable of producing all biomass components above a specified PM threshold. The threshold was either PM >0.1 (solid purple) or PM >0.6 (dashed purple).
DOI: https://doi.org/10.7554/eLife.39733.008

The following figure supplement is available for figure 3:

**Figure supplement 1.** *E. coli* auxotroph co-cultures metabolite producibility.
DOI: https://doi.org/10.7554/eLife.39733.009

our method (the PM for any metabolite) is different from that of FBA (the flux through all reactions), one can use the PM values observed across all biomass components as a proxy for the growth capacity of an organism, providing a metric that is comparable with the FBA-predicted biomass production flux. The specific metrics used to compare the PM and FBA predictions for biomass production are described further in the *Figure 3* legend. One can see that both the FBA and the PM predictions become worse as the metabolic networks are further perturbed. However, the PM predictions are more tolerant to missing reactions than the FBA predictions. While the FBA production of biomass becomes infeasible for the majority of the perturbed metabolic networks after removing less than 1% of the reactions, the PM results remain fairly quantitatively accurate when removing up to 10% of the reactions. This analysis provides insight into the accuracy of our method for analyzing metabolic networks with gaps, such as draft (non-gap-filled) metabolic networks produced through automated reconstruction pipelines.

## Metabolite producibility points to putative metabolic mechanisms for *E. coli* auxotroph co-cultures

As a first test of our approach in its capacity to provide metabolic insight about experimental measurements of inter-microbial interactions, we used the PM to estimate the capacity of different *E. coli*

auxotrophs to compensate for each other's metabolic limitations. In particular, we compared experimental data from co-cultures of *E. coli* auxotrophs from *Wintermute and Silver (2010)* with corresponding PM calculations. After reconstructing in silico the specific auxotroph strains used in this work (based on the *E. coli* iJO1366 metabolic network), we calculated the PM for all essential biomass components in each auxotroph and compared the PM values to the experimentally measured synergistic growth of auxotroph pairs (*Figure 3—figure supplement 1*). Different auxotrophs, clustered by PM, were seen to group based on the pathway of the removed gene, and auxotrophs with knockouts in different locations of the same biosynthetic pathway showed a graded decrease in PM for the corresponding biomass component, similar to what was seen in our histidine biosynthetic pathway analysis in *Figure 2—figure supplement 2*. The overall distance between auxotroph PM values was positively correlated with synergistic growth, suggesting that auxotrophs with different biosynthetic capabilities could better support each other's growth (*Figure 3—figure supplement 1A*). Several examples and counter-examples that further elaborate this trend are highlighted in *Figure 3—figure supplement 1B and C*. This analysis also gave us the opportunity to query in more depth the capacity of our approach to provide insight into whether auxotrophs with higher PM values may be more easily supplemented by partner auxotrophs. We did not detect a clear general signal on whether auxotrophs could rescue each other based on the average PM across all biomass components. However, for a specific instance, namely auxotrophs for tryptophan, we found a correlation between tryptophan PM and average synergistic growth with other auxotrophs (*Figure 3—figure supplement 1D*), possibly suggesting that the PM captures the ease with which auxotrophs in this pathway can be supplemented by other auxotrophs. Overall, our method enables the comparison of model-based producibility predictions with experimental data on auxotrophic interdependencies. These predictions helped identify metabolic complementarity patterns, but did not fully capture all of the complexity of interactions between *E. coli* auxotrophs.

## Reconstruction of human oral microbiome metabolic networks

We next applied our method to the human oral microbiome, aiming at a mechanistic characterization of the biochemical capabilities of different microbes in this community based on metabolic networks reconstructed directly from their genomes. As a first step, we reconstructed metabolic networks for 456 different microbial strains representing a diverse set of human oral microbes whose annotated genomes were available from the Human Oral Microbiome Database. These organisms represent 371 different species, 124 genera, 64 families, 35 orders, 22 classes, and 12 phyla. Metadata related to the selected organisms can be found in *Supplementary file 4*. Notably, the database includes several sequenced yet uncultivated or recently co-cultivated organisms. In particular, the following sequenced yet uncultivated, or recently co-cultivated, strains were included in our analysis: *Saccharibacteria* (TM7) bacterium HMT-952 strain TM7x (*He et al., 2015*), *Saccharibacteria* (TM7) bacterium HMT-955 strain PM004 (*Collins et al., 2019b*), *Saccharibacteria* (TM7) bacterium HMT-488 strain AC001 (*Collins et al., 2019a*), *Tannerella* HMT-286 strain W11667 (*Vartoukian et al., 2016a*), *Anaerolineae* (*Chloroflexi* phylum) bacterium HMT-439 strain Chl2 (*Vartoukian et al., 2016b*), *Absconditabacteria* (SR1) bacterium HMT-874 strain MGEHA (*Campbell et al., 2013a*), and *Desulfobulbus* HMT-041 strains Dsb2 and Dsb3 (*Campbell et al., 2013b*). All of the selected genomes were used to reconstruct sequence-specific draft metabolic networks using the Department of Energy Systems Biology Knowledgebase (KBase) 'build metabolic model' app (*Henry et al., 2010*; *Arkin et al., 2018*; *Overbeek et al., 2014*). The networks were reconstructed without any gap-filling. A KBase narrative containing the genomes and draft metabolic network reconstructions can be found at: https://narrative.kbase.us/narrative/ws.27853.obj.935. The complete collection of all network models is also available for download in MATLAB (.mat) format at https://github.com/segrelab/biosynthetic_network_robustness (*Bernstein, 2019*).

## Large-scale patterns in biosynthetic capabilities identified across the human oral microbiome

We analyzed the PM for 88 different biomass metabolites across the aforementioned 456 metabolic networks from the human oral microbiome. The 88 biomass metabolites included all biomass building blocks considered to be essential for either Gram-negative or Gram-positive biomass, as listed in the KBase build metabolic models app (*Henry et al., 2010*; *Arkin et al., 2018*; *Overbeek et al.,*

*2014*) (listed in *Supplementary file 5*). Through this analysis we calculated 40,128 PM values which represent an atlas of predicted biosynthetic capabilities across these human oral microbiome organisms. The ensuing atlas is represented as a hierarchically clustered matrix of PM values for all 456 organisms and 88 metabolites in *Figure 4*. The same data are available in *Figure 4—figure supplement 1* (clustered by taxonomy), and in *Supplementary file 6*.

The hierarchically clustered heat map (*Figure 4*) shows extensive variability in the PM values of different organisms and metabolites across the oral microbiome. There are three main large clusters of metabolites: one cluster with consistently high PM (top), one cluster with low PM (middle), and one cluster with variable PM (bottom). Different classes of metabolites cluster quite differently across this landscape. In addition to simple ubiquitous metabolites, such as $H_2O$ or glycine (*Figure 4 I*), all nucleotides have high PM across the oral microbiome organisms. Amino acids generally have high PM as well, with the notable exception of tryptophan (*Figure 4 II*). Interestingly, tryptophan is known to be a particularly difficult amino acid to synthesize (*Akashi and Gojobori, 2002*). Metal ions generally had PM value of 0 across all organisms, serving as an expected negative control. Some exceptions, such as $Mg^{2+}$, $Co^{2+}$, $Cl^-$, $Fe^{3+}$, and $Fe^{2+}$, can be explained based on their presence in larger compounds, such as porphyrins. For example, $Co^{2+}$ has increased PM values in a pattern that closely follows the PM values of the cobalt containing vitamin cobamamide (*Figure 4 III*).

Before analyzing in detail the patterns identifiable in the PM matrix of *Figure 4*, we showed that such patterns could not be simply attributed to the broad property of genome size – even if genome size is known to be an important predictor of the overall biosynthetic capabilities of an organism (*Zarecki et al., 2014*). Fastidious or parasitic organisms tend to have reduced genomes and consequently reduced metabolic capabilities. In our data, the overall average PM value for each organism can be partially predicted by genome size. A linear regression model and quadratic regression model which used the log of genome size to predict the average PM value across all metabolites for each organism had R-squared values of 0.498 and 0.551 respectively (*Figure 4—figure supplement 2 A*). The fit of this model was further improved by adding taxonomic information as additional parameters (see methods section for additional details on adding taxonomic information). We inferred this by using the Akaike information criterion (AIC) and Bayesian information criterion (BIC), two measures of model accuracy that include a penalty for added parameters to discourage over-fitting (*Clarke et al., 2009*). The BIC has a stronger penalty for additional parameters and improved up to the order level, while the AIC improved up to the genus level (*Figure 4—figure supplement 2 B, C*). These improvements in AIC and BIC indicate that our data contain additional structure that is described by taxonomy beyond simply genome size.

## Taxonomic trends capture biosynthetic patterns across human oral microbiome organisms

Many of the patterns in our large-scale analysis of the human oral microbiome PM matrix indicated taxonomic trends in the PM of different metabolites across organisms. While the clustering of the PM matrix was not entirely driven by taxonomy (*Figure 4*), we did see significant taxonomic trends in our data beyond what was explained simply be genome size (*Figure 4—figure supplement 2*). We further investigated, quantitatively, which specific phyla and orders were associated with specific PM trends by calculating the log likelihood ratio between a quadratic regression model predicting the PM values for a particular metabolite-based solely on genome size against one that incorporates a specific taxonomic parameter of interest (*Figure 4—figure supplement 3*). This allowed us to highlight metabolites with highly significant increased or decreased PM values in certain taxonomic groups, and to confirm patterns that we observed by eye in *Figure 4*. Numerous patterns and details of the PM matrix could be relevant for addressing specific biological questions or model refinement challenges. Here we focus in detail on two specific classes of compounds: (i) cell-wall and membrane components, which tend to vary broadly across organisms, and are important for antimicrobial susceptibility and immune system recognition; and (ii) amino acids and essential factors (e.g. vitamins), which could be relevant for understanding metabolic exchange among bacteria and with the host.

A first striking pattern in the PM matrix is the complexity of cell-wall and membrane components of different taxa. Some aspects of this pattern are consistent with standard attribution of metabolites associated with the Gram staining categories (estimated using the KBase build metabolic model app [*Henry et al., 2010*; *Arkin et al., 2018*; *Overbeek et al., 2014*]). However, we also observed

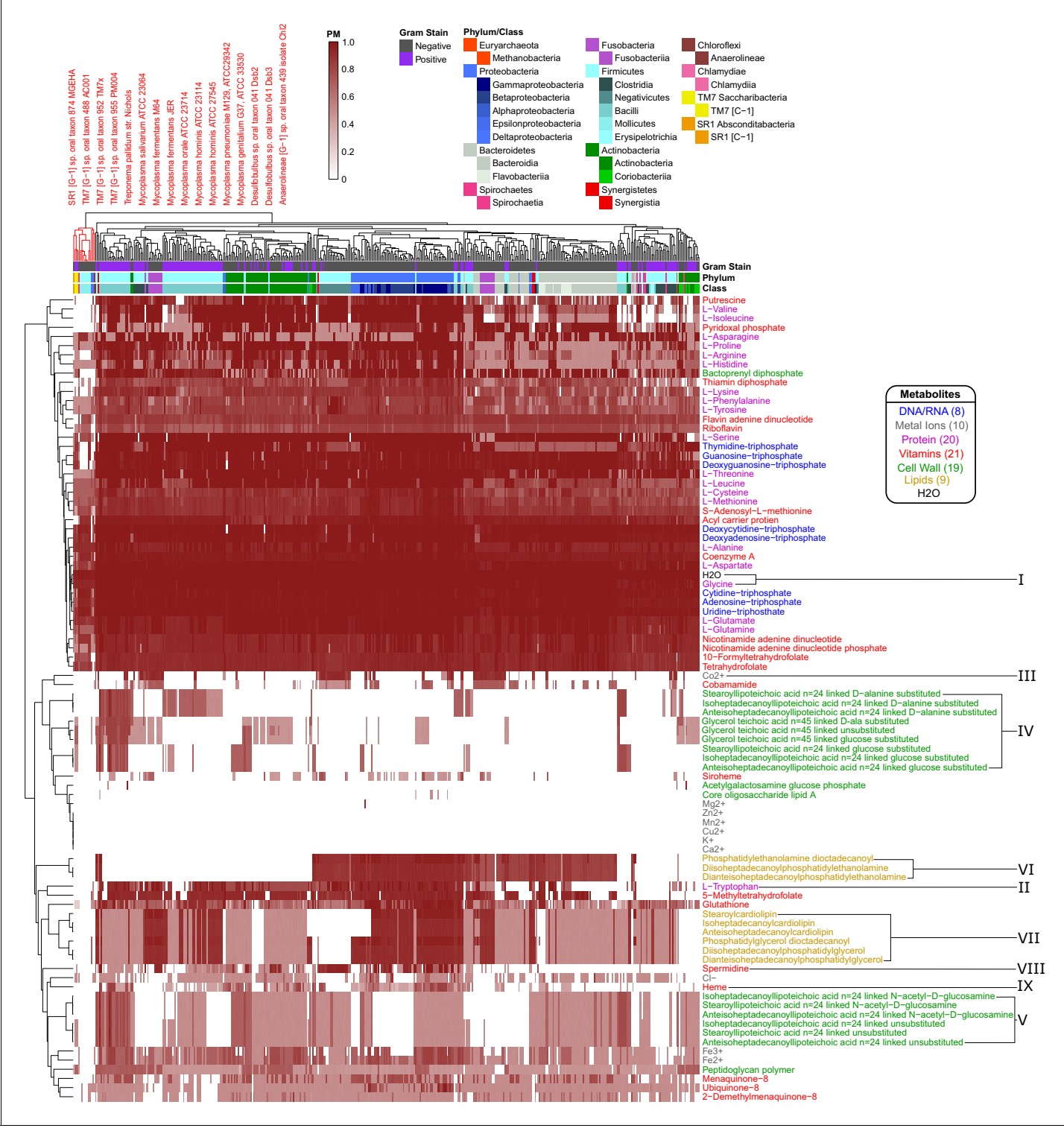

**Figure 4.** Human oral microbiome organisms PM matrix. The producibility metric (PM) was calculated for 456 different oral microbiome organisms (columns) and 88 different essential biomass metabolites (rows). The resulting matrix is hierarchically clustered based on average distances between organisms and metabolites PM values. Organism Gram-stain and phylum/class are indicated by several annotation columns at the top of the matrix. The biomass metabolites analyzed consisted of several different types of metabolites indicated with different colors. Several metabolites that showed interesting patterns across oral microbiome organisms are highlighted with roman numerals. The most distinct cluster of organisms, highlighted and annotated (top left), consisted of fastidious reduced-genome organisms (*Mycoplasma*, *Treponema*) and uncultivated or recently cultivated organisms (*SR1*, *TM7*, *Desulfobulbus*, *Anaerolineae*).

*Figure 4 continued on next page*

*Figure 4 continued*

DOI: https://doi.org/10.7554/eLife.39733.010

The following figure supplements are available for figure 4:

**Figure supplement 1.** Taxonomically ordered human oral microbiome organisms PM matrix.
DOI: https://doi.org/10.7554/eLife.39733.011

**Figure supplement 2.** Prediction of average producibility metric using genome size and taxonomic parameters.
DOI: https://doi.org/10.7554/eLife.39733.012

**Figure supplement 3.** Taxonomic parameters as predictors of metabolite specific producibility.
DOI: https://doi.org/10.7554/eLife.39733.013

**Figure supplement 4.** Producibility of different organic acids across human oral microbiome organisms.
DOI: https://doi.org/10.7554/eLife.39733.014

**Figure supplement 5.** Analysis of metabolite producibility change for a proteolytic organism (*Porphyromonas gingivalis*) and a saccharolytic organism (*Streptococcus mutans*) in a protein enriched environment.
DOI: https://doi.org/10.7554/eLife.39733.015

**Figure supplement 6.** Correlations of various pairwise metabolic metrics.
DOI: https://doi.org/10.7554/eLife.39733.016

**Figure supplement 7.** Comparison of PM complementarity vs Seed complementarity.
DOI: https://doi.org/10.7554/eLife.39733.017

interesting deviations, which could be partially attributed to known finer resolution in the specific membrane components across taxa. Compared to other metabolites, cell-wall components generally tend to have variable or low PM values across the oral microbiome organisms. We analyzed in detail fifteen different teichoic acids, a class of metabolites expected to be found in the cell wall of Gram-positive organisms that play an important role in microbial physiology and interactions with the host (***Weidenmaier and Peschel, 2008***). Of these, nine were found to have higher PM values in Gram-positive organisms, as expected (***Figure 4 IV***). In particular, the D-alanine substituted lipoteichoic acids had high PM values in the phylum *Firmicutes* and specifically the class *Bacilli*. However, there was another set of 6 teichoic acids that had intermediate PM values across a large number of organisms and didn't follow Gram-staining trends (***Figure 4 V***). These consisted of three N-acetyl-D-glucosamine linked and three unsubstituted teichoic acids. This mismatch in expected patterns suggests that the metabolic pathways involving these particular cell-wall components may merit closer inspection in the network reconstruction process.

We further observed clear trends associated with several lipids which are expected to be found in the cell membrane of both Gram-positive and Gram-negative organisms. In particular, we found a strong increase in the PM value for three phosphatidylethanolamine lipids in Gram-negative organisms (***Figure 4 VI***). Interestingly, these lipids have been previously observed to be more commonly produced in Gram-negative organisms, and have implications for antimicrobial susceptibility (***Epand et al., 2007***; ***Epand and Epand, 2009***). We also identified trends associated with three cardiolipin and three phosphatidylglycerol lipids that display generally similar PM patterns across different species (***Figure 4 VII***). One class of organisms that stands out with respect to lipid biosynthesis are the *Negativicutes*. These organisms have relatively high PM values for phosphatidylethanolamine but PM values of 0 for cardiolipin and phosphatidylglycerol lipids. Consistent with this result, it has been previously observed that the *Negativicutes* organism *Selenomonas ruminantium* lacks cardiolipin and phosphatidylglycerol lipids in its inner and outer cell membranes, but does have phosphatidylethanolamine (***Kamio and Takahashi, 1980***). It has been hypothesized that the membrane stabilizing role of these two missing lipids could be partially fulfilled by peptidoglycan bound polyamines, including spermidine, in *Selenomonadales* organisms (***Kamio and Takahashi, 1980***; ***Hamana et al., 2012***). Concordantly, we see an increased PM value for the polyamine spermidine across *Negativicutes* in our data (***Figure 4 VIII***). These patterns suggest that the PM could be used to obtain organism-specific estimates of biomass composition from genomes for metabolic network reconstruction, facilitating assignments beyond gram positive/negative compositions.

Aside from lipids and cell-wall components, there are a number of interesting trends related to several amino acids and other essential factors in our data. A number of metabolites had increased PM in the phylum *Proteobacteria* and decreased PM values in the phylum *Bacteroidetes*. A notable example is heme, which can be seen to follow this trend (***Figure 4 IX***). Heme plays an important role

in microbe host interactions, as bacterial pathogens often acquire it from their human host (*Choby and Skaar, 2016*). In the context of the human oral microbiome, the oral pathogen *Porphyromonas gingivalis* (belonging to the phylum *Bacteroidetes*) is known to scavenge heme (*Olczak et al., 2005*), compatible with the above pattern. Other metabolites that displayed the same trend include: arginine, cysteine, methionine, tryptophan, and glutathione. Arginine can be catabolized via the arginine deiminase pathway to regenerate ATP and is thus an interesting exchange metabolite beyond its use as a protein building block (*Plugge and Stams, 2001*; *Schink, 2006*). Tryptophan is one of the highest cost amino acids to biosynthesize (*Akashi and Gojobori, 2002*), and thus is an intriguing exchange candidate. Methionine and Cysteine are the only two sulfur containing standard amino acids, and glutathione is synthesized from Cysteine. It is possible that the discrepancies between PM values observed here are indicative of broad amino acid and vitamin exchange between the phyla *Proteobacteria* and *Bacteroidetes* in the human oral microbiome.

## Organic acid production predicted for human oral microbiome organisms

In addition to calculating the producibility of biomass components, we were interested in applying the PM to other metabolites that could be produced by microbes and impact microbial community structure or function in the human oral microbiome. We thus used our method to compute the PM of various organic acids across oral microbiome organisms. We analyzed nine different organic acids and observed a large amount of variability in PM (*Figure 4—figure supplement 4*). Acetate had the highest median PM while butyrate had variable PM, with most organisms having PM of 0 but some having relatively high PM. In particular, *Fusobacterium* genus organisms were found to have high PM for butyrate, reflecting observations obtained from transcriptomic data, with important implications for periodontal disease (*Jorth et al., 2014*). Additionally, increased butyrate PM was observed in some but not all *Porphyromonas* and *Prevotella* species, which have been further implicated in periodontal disease due to their potential production of inflammation inducing organic acids (*Takahashi, 2015*). For reference, the organic acids analyzed in this section were added to *Supplementary file 5*, and the calculated PMs were added to *Supplementary file 6*.

## Metabolite producibility in a protein vs. carbohydrate-enriched environment

Although one of the most useful features of our method is the capacity to provide an environment-independent measure of metabolite producibility, it can also be tailored to ask environment-specific questions. To exemplify this capability, we applied our method to investigate the biosynthetic capabilities of a proteolytic organism (*Porphyromonas gingivalis*) and a saccharolytic organism (*Streptococcus mutans*) in a protein and a carbohydrate-enriched environment. The hypothesis was that the proteolytic organism would have a higher PM increase in the protein enriched environment as it is able to breakdown amino acids to synthesize other biomass components and likewise the saccharolytic organism would have a higher PM increase in the carbohydrate-enriched environment. We simulated a protein-enriched environment by fixing all amino acids to always be present ($P_{in}$ = 1) when calculating the PM, and simulated a carbohydrate-enriched environment by fixing D-glucose to always be present ($P_{in}$ = 1). Target metabolites were never fixed to be present; for example when calculating the PM of an amino acid in the protein-enriched environment we did not fix that amino acid to be present. We measured the increase in PM in the enriched environments relative to the originally calculated PM, for all 88 biomass metabolites and nine organic acids (*Figure 4—figure supplement 5*). Overall, we saw only small increases in PM in the enriched environments, with particularly small increases in the carbohydrate-enriched environment. The modest trends that we identified matched our expectation, with the proteolytic organism showing a larger increase in PM in the protein-rich environment and the saccharolytic organism showing a larger increase in PM in the carbohydrate-enriched environment. One possible reason for the small effects observed in this analysis is the fact that our baseline random environment is fairly rich. For example, fixing D-glucose to be available in the carbohydrate-enriched environment had minimal effect as D-glucose already had a high PM in the original random environment. However, this application does highlight the value of

further exploring variants of our method that explicitly translate environmental information into non-uniform metabolite input probabilities.

## Metabolic similarity correlates with microbial co-occurrence in the human oral microbiome

Our approach is a bottom-up approach, that starts from genomes and predicts metabolic capabilities that could underlie interactions. A key question in the field of metabolic modeling is whether these bottom-up metrics can be compared to and provide insight into top-down analyses of large datasets, such as the patterns of co-occurrence of microbial taxa from microbiome sequencing data. To address this question for our approach, we used the PM to calculate pairwise measures of metabolic difference or complementarity between any two organisms and assessed the correlation of these metrics with microbial co-occurrence. Through this analysis we sought to identify metabolic trends associated with co-occurring microbes. We simultaneously evaluated the correlation between microbial co-occurrence and other previously defined metrics (*Carr and Borenstein, 2012*; *Kreimer et al., 2012*; *Levy et al., 2015*), so that we could compare these to the performance of the PM. While there are a few additional methods that have utilized gap-filled metabolic models to provide insight into microbial co-occurrence data (*Freilich et al., 2011*; *Zelezniak et al., 2015*), in this study we focused our direct comparison on alternative methods that could be used to analyze draft (non-gap-filled) metabolic networks as these were closest in scope and applicability to our own method. Future analyses could broaden the scope of this comparison. All of the pairwise metabolic metrics we calculated are described further in the methods section. For co-occurrence data, we analyzed the supplementary data from *Friedman and Alm (2012)*, which contains microbial co-occurrences identified from 16S rRNA sequencing data using their SparCC method for seven different oral microbiome sites. The correlations between all pairwise metabolic metrics and microbial co-occurrence in all seven oral microbiome sites are presented in *Supplementary file 7*.

Across the seven different oral microbiome sites, the pairwise metabolic metric 'PM distance' (see methods section for description of metrics) showed the most consistent significant correlation with co-occurrence of any pairwise metabolic metric. The PM distance was consistently negatively correlated with the co-occurrence, indicating that organisms that are more similar in PM tend to co-occur. Several other pairwise metabolic metrics were found to be correlated with co-occurrence, although in a less consistent manner than the PM distance (*Supplementary file 7*). Additionally, many of the pairwise metabolic metrics that we analyzed were highly correlated with each other as we show in *Figure 4—figure supplement 6*. To further disentangle correlations between pairwise metabolic metrics and co-occurrence data, we looked at the partial-correlation between a pairwise metabolic metric and co-occurrence when controlling for another pairwise metabolic metric. We found that the PM distance always had significant partial-correlation with co-occurrence when controlling for any of the other pairwise metabolic network metrics, a trend not observed for the other metrics. We further repeated this entire correlation analysis for co-occurrence measured by Pearson's correlation (also from the supplementary data of *Friedman and Alm, 2012*.), and interestingly found that correlations between pairwise metabolic metrics and co-occurrence were weaker and less consistent when using Pearson's correlation, in line with previously reported inconsistency in co-occurrence prediction by Pearson's correlation (*Friedman and Alm, 2012*). Overall, our analysis corroborates and enhances previous analyses showing how co-occurrences in 16S rRNA sequencing data from the human microbiome project tend to reflect 'habitat filtering', where organisms with similar metabolic capabilities tend to co-occur (*Freilich et al., 2011*; *Zelezniak et al., 2015*; *Levy and Borenstein, 2013*).

We next examined more closely the correlation between the pairwise metabolic metrics PM complementarity and Seed complementarity (see Methods and *Figure 4—figure supplement 7*). These measures of complementarity summarize the potential for any one organism to provide metabolic products to another. While the two metrics are highly correlated with each other, the distribution of their values display some significant differences. In particular, the PM complementarity displays a clear bi-modal distribution, which is absent from the distribution of Seed complementarity values. The high-valued peak of the PM complementarity distribution captures most of the interactions between small-genome/fastidious microorganisms and their partners. This indicates that our method is good at resolving biosynthetic deficiencies in fastidious/uncultivated organisms, as further investigated next.

## Biosynthetic properties predicted in a cluster of fastidious human oral microbiome organisms

In addition to dissecting the patterns associated with specific metabolites, one can analyze the PM landscape of *Figure 4* from the perspective of the organisms and their agglomeration into clusters. Strikingly, in our large clustered PM matrix, the most distinct hierarchical cluster of organisms consisted of a number of fastidious organisms (*Figure 4* top left). This cluster included all of the *Mycoplasma* genomes that we analyzed, and one *Treponema* genome. *Mycoplasma* and *Treponema* are genera that are known to be parasitic and have evolved to have reduced genomes and metabolic capabilities (*Fraser et al., 1995*; *Fraser et al., 1998*; *Meseguer et al., 2003*; *Davis et al., 2013*; *Razin, 1978*). The remaining members of this cluster included nearly all of the sequenced yet uncultivated, or recently co-cultivated, organisms in our study. The organisms included were from the phyla: *Absconditabacteria* (SR1), *Saccharibacteria* (TM7), *Proteobacteria* (genus *Desulfobulbus*), and *Chloroflexi* (class *Anaerolineae*). Many of these organisms are thought to have reduced genomes and limited metabolic capabilities underlying their fastidious nature, much like *Mycoplasma.* Only one of the previously uncultivated organism we analyzed was found outside of this fastidious cluster, namely *Tannerella* HMT-286. Interestingly, this bacterium is hypothesized to rely on externally supplied siderophores to support its growth (*Vartoukian et al., 2016a*). This type of dependency is not captured by our metabolic analysis and highlights the fact that, while uncultivability can be driven by many different mechanisms, our method captures the prominent effect of reduced biosynthetic capacity.

We sought to gain clearer insight into the metabolic properties of these co-clustered fastidious organisms by re-clustering their PM submatrix (*Figure 5 A*). By comparing the PM values in this fastidious cluster to those in the average oral microbiome organisms, it is clear that the fastidious organisms had reduced PM for a large number of metabolites including cell-wall components, lipids, amino acids, and other essential factors. When ranking metabolites by their difference in average PM between all oral microbiome organisms and the fastidious cluster a number of amino acids and vitamins stand out as being the most depleted in the fastidious cluster. The top metabolites where: pyridoxal phosphate, valine, putrescine, isoleucine, bactoprenyl diphosphate, thiamin diphosphate, 5-methyltetrahydrofolate, lysine, deoxyguanosine triphosphate, tryptophan, and guanosine-triphosphate. These metabolites may be particularly relevant with regards to exchange between fastidious organisms and their oral microbiome community partners. Amino acids, in particular, have been hypothesized to be involved in metabolic exchange between microbial organisms in communities (*Ponomarova and Patil, 2015*; *Mee et al., 2014*; *Mee and Wang, 2012*; *Zelezniak et al., 2015*). Amino acids with reduced PM in the fastidious cluster tend to have high biosynthetic cost (cost calculated in *Akashi and Gojobori, 2002*.), as indicated by Spearman correlation analysis ($\rho = 0.4595$, p-value=0.0415). An exception to this trend, potentially interesting for follow up studies, is the case of the branched chain amino acids valine, and isoleucine, which are the two amino acids with most reduced PM in fastidious organisms, but are not among the costliest. Notably, branched chain amino acid supplementation has been shown to alter the metabolic structure of the gut microbiome of mice (*Yang et al., 2016*).

We next sought to gain more specific insight into a specific class of recently-cultivated fastidious organisms, *Saccharibacteria* (TM7). To gain specific insight into the biosynthetic capabilities of these TM7 relative to other fastidious microorganisms, we further focused our analysis on identifying discrepancies between *Mycoplasma* and TM7. Our analysis included eight *Mycoplasma* genomes and three TM7 genomes. *Mycoplasma* are a relatively well characterized genus of intracellular parasites with reduced metabolic capabilities, and TM7 are a recently co-cultivated phylum of the candidate phyla radiation that display reduced metabolic capabilities and a parasitic lifestyle. There are several cell-wall components for which TM7 has relatively high PM values and Mycoplasma has PM values of zero (*Figure 5 I*). These include nine different teichoic acids, bactoprenyl diphosphate, and peptidoglycan. This highlights extensive cell-wall/peptidoglycan metabolism in TM7 organisms and the known lack of a cell-wall in *Mycoplasma* (*Razin, 1978*). Furthermore, a set of three nucleotides: dGTP, GTP, and TTP, have high PM values for TM7 and PM values of zero for *Mycoplasma* organisms (*Figure 5 II*). This pattern of nucleotide biosynthesis deficiency in *Mycoplasma* is consistent with the observation that some strains have been shown to be dependent on supplementation of thymidine and guanosine but not adenine or cytosine nucleobases for growth (*Mitchell and Finch, 1977*).

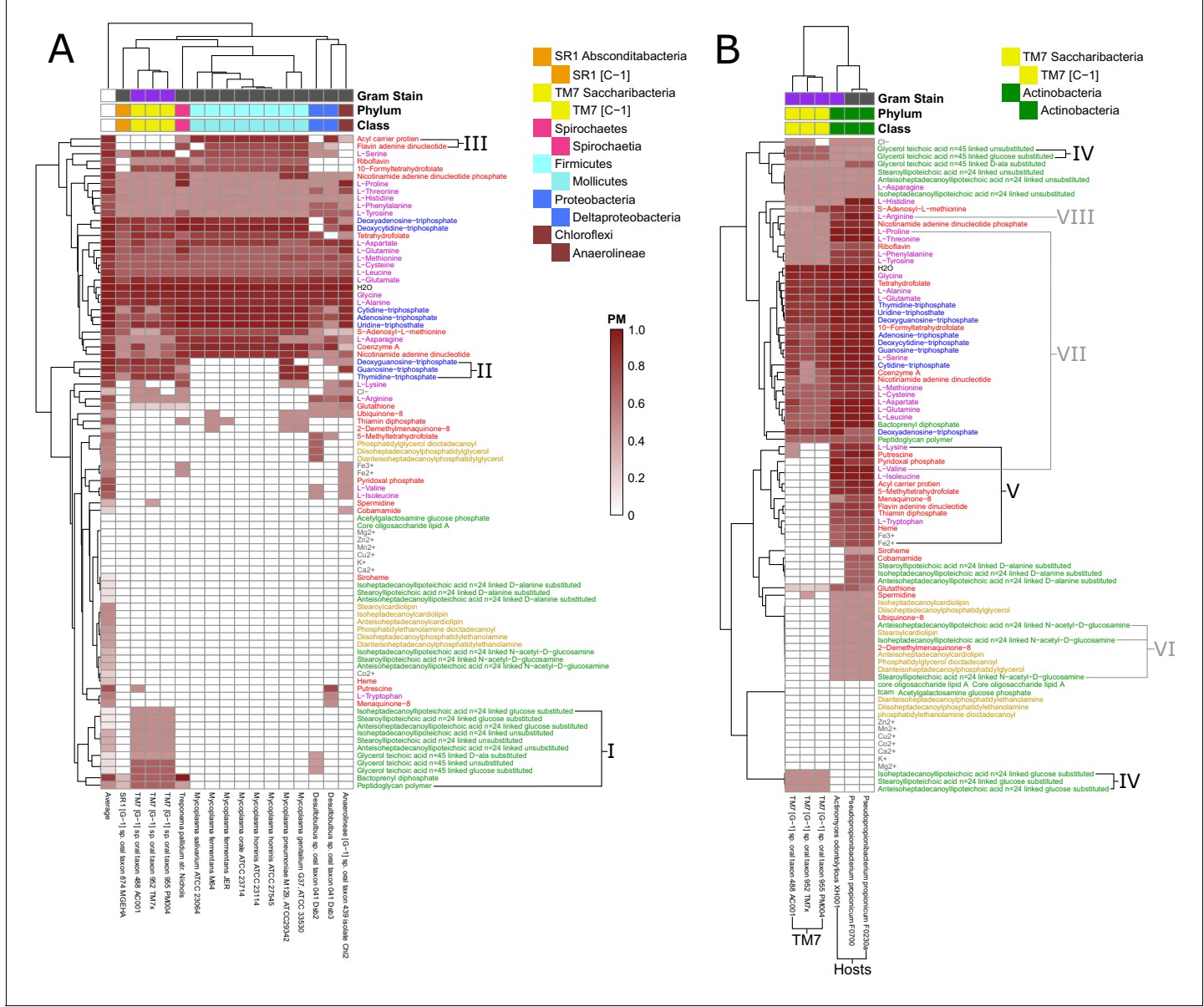

**Figure 5.** Fastidious/uncultivated and TM7/host producibility sub-matrices. Sub-matrices of the larger PM matrix were re-clustered to highlight variations within specific groups of fastidious and uncultivated organisms. (A) The fastidious/uncultivated organisms that were identified as the most unique cluster in the larger matrix from *Figure 4* were re-clustered hierarchically. The average producibility metric (PM) value across all oral microbiome organisms analyzed in this study is shown in the far left column. Differences between the fastidious *Mycoplasma* genus organisms and the previously uncultivated TM7 organisms are highlighted with roman numerals. (B) The PM values for the previously uncultivated TM7 organisms and their growth-supporting hosts bacteria were extracted and re-clustered hierarchically. Differences between the TM7 and their bacterial hosts are highlighted with roman numerals.

DOI: https://doi.org/10.7554/eLife.39733.018

Finally, the cofactors acyl carrier protein (ACP) and flavin adenine dinucleotide (FAD) had high PM values in *Mycoplasma* and PM values of zero in TM7 organisms (*Figure 5 III*). The lack of these cofactors in TM7 seems surprising, but is indeed matched by a complete lack of any metabolic reactions annotated to utilize FAD and ACP as cofactors in the draft reconstruction of the TM7 metabolic networks.

In addition to investigating the metabolic deficiencies of fastidious organisms, the PM landscape gave us the opportunity to compare these gaps with possible complementary capabilities in

organisms known to support their growth. The three TM7 strains that we analyzed were recently co-cultivated with host bacteria from the human oral microbiome. TM7x was shown to be a parasitic epibiont of *Actinomyces odontolyticus* XH001 (*McLean et al., 2016*). TM7 AC001 and PM004 were recently both co-cultivated successfully with either of the host strains *Pseudopropionibacterium propionicum* F0230a or F0700 (*Collins et al., 2019c*). We further investigated these newly discovered relationships to gain insight into possible metabolic exchange (*Figure 5 B*). Interestingly, TM7 organisms had higher PM values than their host strains for several cell-wall components: three glucose-substituted teichoic acids, and glucose-substituted and unsubstituted glycerol teichoic acid (*Figure 5 IV*), suggesting that TM7 is capable of producing several cell-wall components that its host cannot. Conversely, as expected, a large number of metabolites had increased PM values in the host strains compared to the TM7 strains. These metabolites are hypothesized to be easily synthesized by the host and not TM7 and are thus interesting candidates for growth supporting exchange. Fourteen different metabolites had average PM values in the hosts greater than 0.60 higher than in the TM7 organisms (*Figure 5 V*). The ranked list includes: isoleucine, valine, acyl carrier protein, 5-methyltetrahydrofolate, pyridoxal phosphate, flavin adenine dinucleotide, thiamin diphopsphate, putrescine, tryptophan, $Fe^{2+}$, heme, $Fe^{3+}$, lysine, and menaquinone-8. Interestingly, the branched chain amino acids isoleucine and valine are again at the top of the list. The correlation of amino acid biosynthesis cost (*Akashi and Gojobori, 2002*) with the difference in PM values between host and TM7 is even higher than what we observed across all fastidious organisms (Spearman correlation $\rho = 0.6011$, p-value=0.0051) indicating that PM values are further decreased in TM7 for costly amino acids.

Our results provide context and putative mechanistic details related to observed gene expression and metabolic changes in the co-cultivation of TM7x with the host *Actinomyces odontolyticus* XH001 (*McLean et al., 2016*). Transcriptomic data for TM7x and *A. odontolyticus* XH001 showed that a number of genes associated with N-acetyl-D-glucosamine were up regulated in *A. odontolyticus* in this interaction (*He et al., 2015*). Our results show that, although TM7 does have extensive cell wall metabolism, *A. odontolyticus* has higher PM for N-acetyl-D-glucosamine substituted components (*Figure 5 VI*). This suggests that the host is responsible for the biosynthesis of these cell-wall components, which may be overexpressed during co-cultivation. Metabolomics experiments from this co-cultivation have identified the cyclic peptide cyclo(L-Pro-L-Val) as a potential signaling molecule in this relationship. Our PM analysis suggests that this molecule would be synthesized by the host as it has increased PM values for both of the amino acids included (*Figure 5 VII*). In fact, valine has one of the highest discrepancies in PM for host and TM7. Finally, another potentially exchanged amino acid of interest is arginine. All three TM7 draft metabolic network reconstructions that we analyzed were annotated to possess either all or all but one of the reactions in the arginine deiminase pathway (TM7 PM004 is missing the arginine iminohydrolase reaction) (*Figure 6—figure supplement 1* and Supplementary Cytoscape (*Shannon et al., 2003*) files 1–3). This catabolic pathway can be used to degrade arginine to regenerate ATP, and has been implicated in syntrophic microbial interactions (*Plugge and Stams, 2001*; *Schink, 2006*). In our PM analysis arginine had consistently higher PM in host than TM7 (*Figure 5 VIII*). Thus, arginine exchange and metabolism via the arginine deiminase pathway could contribute to the dependence of TM7 on its hosts (*Figure 6*).

## Discussion

Our method provides an estimate of the putative biosynthetic capabilities of a metabolic network from genomic information. We first implemented this method in *E. coli*, to demonstrate its application and capacity to address multiple questions, even in presence of uncertainty that would prevent the use of other stoichiometric methods. Next, we reconstructed metabolic networks for 456 different organisms from the human oral microbiome, and generated an atlas of predicted biosynthetic capabilities across these organisms. We highlighted trends in the biosynthetic capabilities of these microbes related to taxonomy, and showed that these predicted biosynthetic capabilities can partially explain co-occurrence data. We further focused on describing putative biosynthetic deficiencies of a cluster of fastidious/uncultivated organisms and predicted exchanged metabolites between three recently co-cultivated *Saccharibacteria* (TM7) strains and their growth supporting partner microbes. Overall, our method provides preliminary insight into the metabolic capabilities of a large

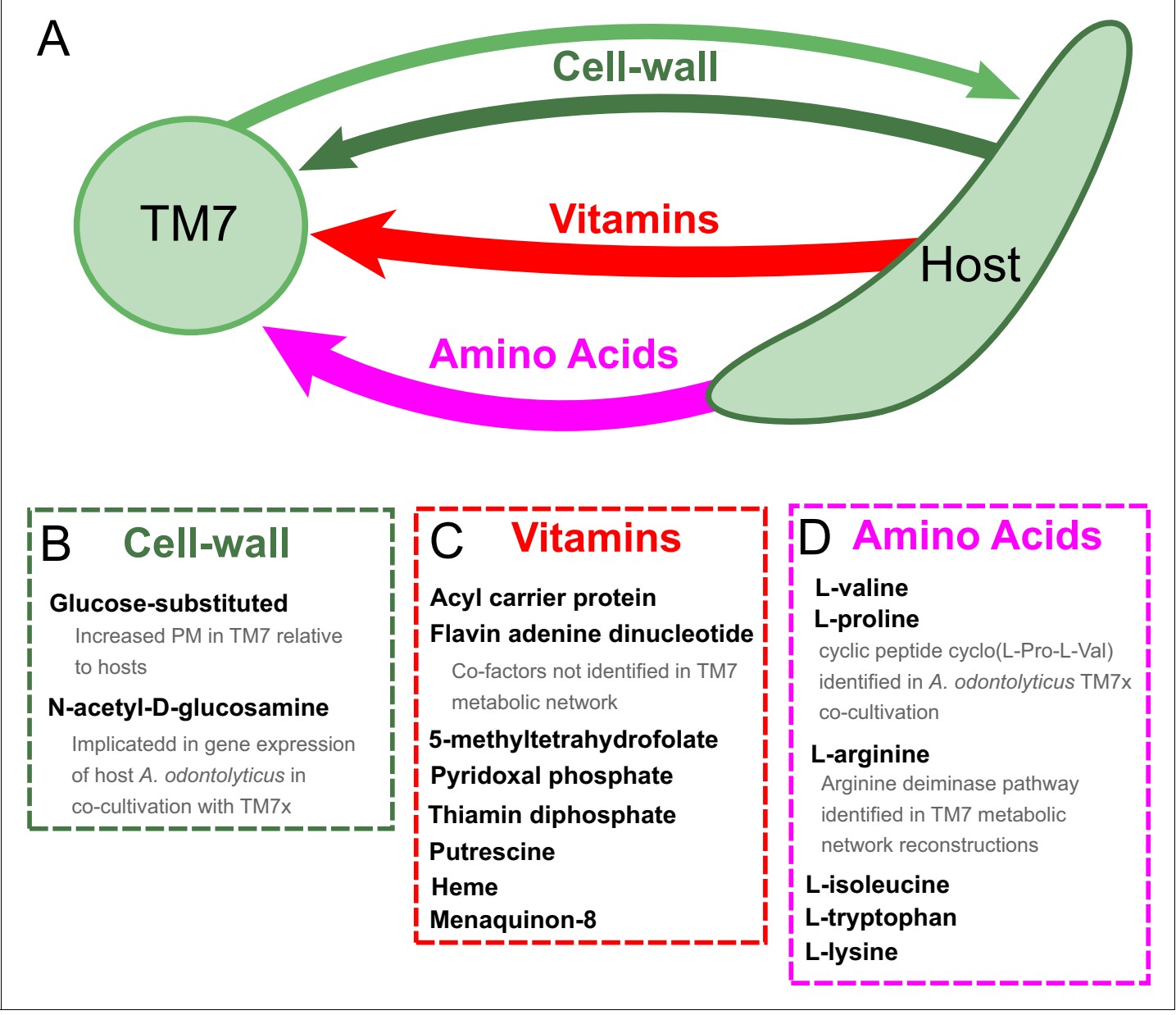

**Figure 6.** Hypothesized metabolic exchange between TM7 and their bacterial hosts. (**A**) We summarize here hypotheses generated for the exchange of metabolites between TM7 and their growth-supporting hosts based on differences in biomass PM values. We also highlight any insight that our PM was capable of providing into experimental transcriptomic and metabolomic data from the co-cultivation of TM7x and *Actinomyces odontolyticus* that was previously collected and analyzed in a separate study (*He et al., 2015*). (**B**) The cell-wall components containing glucose-substituted teichoic acids were among the only metabolites with PM higher in TM7 than in hosts. N-acetyl-D-glucosamine-substituted teichoic acids had increased PM in the host relative to TM7, and previous gene expression data from TM7x and *A. odontolyticus* shows several genes related to N-acetyl-D-glucosamine that are overexpressed in *A. odontolyticus* during co-cultivation (*He et al., 2015*). (**C**) Several vitamins/cofactors/other essential factors had decreased PM in TM7 compared to the hosts. The cofactors acyl carrier protein and flavin adenine dinucleotide had decreased PM in TM7, and were also not found to be utilized in the TM7 draft metabolic network reconstructions. (**D**) Several amino acids had decreased PM in TM7 compared to the hosts. Valine and proline were both decreased in TM7 relative to the host, and previous metabolomics data from TM7x and *A. odontolyticus* identified the cyclic dipeptide cyclo(L-Pro-L-Val) as a potential signaling molecule (*He et al., 2015*). Arginine had decreased PM in TM7 relative to the host and could potentially be exchanged and catabolized by TM7 via the arginine deiminase pathway.

DOI: https://doi.org/10.7554/eLife.39733.019

The following figure supplement is available for figure 6:

**Figure supplement 1.** TM7 metabolic network visualization.

DOI: https://doi.org/10.7554/eLife.39733.020

number of human oral microbiome organisms and helps further the understanding of the structure of this complex microbial ecosystem.

Our method differs from other approaches in several key ways that we have demonstrated throughout our analysis. First, it is different from a pathway-based analysis where the percent completion of biosynthetic pathways is analyzed. Our method is based on the entire metabolic network and captures the multitude of different routes through which a metabolite could be synthesized. An additional advantage is that our method does not rely on previously defined biosynthetic pathway annotations and instead seeks to use the entire metabolic network structure to define biosynthetic capabilities. While prior knowledge of pathways can be quite useful in many contexts, specific biosynthetic routes can cross the boundaries of annotated pathways, making pathway completeness uninformative. A prior notable example of this effect is the discovery of an alternative pathway to bypass a TCA cycle gene impairment (*Frezza et al., 2011*). This pathway connects in an unexpected way two distinct pathways, generating a new crucially important and experimentally validated type of connectivity that would be missed from regular pathway-based analysis. Another such example, in our current data, is the case of the arginine deiminase pathway and the urea cycle, which contain several overlapping metabolites and reactions. In fact, we have noticed that KEGG mappings of TM7x metabolism often highlight the urea cycle as a result of TM7x containing the complete arginine deiminase pathway. Our method differs substantially also from standard flux balance analysis, even if it is based on stoichiometry and Linear Programming. Specifically, our method has improved tolerance for missing reactions compared to flux balance analysis (*Figure 3*), and thus does not rely on gap-filled metabolic networks. Therefore, it is capable of providing preliminary insight into 'draft' genome-derived metabolic networks that can be used to study diverse microbes and microbial communities, and could potentially help guide the gap-filling process and predict putative biomass components. Our method also differs from alternative topology-based methods (*Borenstein et al., 2008*; *Carr and Borenstein, 2012*; *Kreimer et al., 2012*; *Levy et al., 2015*) as it represents metabolism as a bipartite graph constrained by stoichiometry (enabling enforcement of mass balance constraints), rather than projecting the network onto an adjacency matrix between metabolites, which is not constrained by stoichiometry (i.e. two metabolites can be connected in an adjacency matrix despite a missing reactant or cofactor for the reaction that connects them).

It is important to highlight the limitations of our approach. In particular, many of the issues that limit the accuracy of metabolic network analyses in general affect our method as well. The primary limitation is enzyme annotation. Aside from missing or incorrect annotations, subtle processes such as enzyme promiscuity and spontaneous reactions may have unquantified effects on metabolic network function. Reaction direction/reversibility is also difficult to predict as it requires detailed knowledge of reaction thermodynamics and metabolite concentrations. In particular, inaccurate or missing information about reaction direction/reversibility could lead to uncertainty about whether a high PM from our method should be interpreted as reflecting biosynthetic or degradative capabilities (or both). Throughout our analysis we have utilized default reversibility constraints provided by the KBase build metabolic models app (*Henry et al., 2010*; *Arkin et al., 2018*; *Overbeek et al., 2014*), but more stringent constraints on directionality could possibly improve our results. Transport reactions are also notoriously difficult to annotate accurately, and the current implementation of our method addresses this limitation by naïvely adding intracellular metabolites as input metabolites. However, any future efforts to use extracellular metabolites as inputs would rely on accurate transport reaction annotations. In general, all metabolic network analysis methods face similar limitations. Even as newly developed experimental methods gradually improve metabolic reaction annotation (*Price et al., 2018a*; *Vaccaro et al., 2016*; *Price et al., 2018b*; *Sévin et al., 2017*), it is likely that we will have to continue dealing with incomplete knowledge. Thus, approaches like the one presented here are valuable for providing initial predictions of metabolic capabilities with minimal arbitrary assumptions, and for pinpointing specific areas of a metabolic model that are in need of refinement. One additional limitation of our method, in comparison to alternative methods, is that it requires a longer run time than alternative methods, such as FBA (*Orth et al., 2010a*) or NetSeed (*Borenstein et al., 2008*; *Carr and Borenstein, 2012*). Future efforts towards simplifying the calculations to improve the algorithm's speed would be beneficial. For example, utilizing heuristics or belief propagation could possibly improve the efficiency and run time of our algorithm (*Yedidia et al., 2003*).

Despite these limitations, by translating genotype into phenotype with minimal assumptions, our approach has the potential to serve as a baseline estimate of metabolic mechanisms in different microbial communities. Moving forward, our method could be easily applied to other human-associated or environmentally relevant microbial communities, providing valuable putative insight into inter-microbial metabolic dependencies. For example, in this analysis we have analyzed three previously uncultivated *Saccharibacteria* (TM7) phylum organisms that were recently successfully co-cultivated with growth supporting bacterial host organisms. These TM7 species are the first successfully cultured organisms from the candidate phyla radiation, a large branch of the tree of life consisting mainly of uncultivated organisms (*Kantor et al., 2013*; *Brown et al., 2015*; *Hug et al., 2016*), and therefore are of general interest beyond their role in human oral health. Further analysis of the candidate phyla radiation through our method could provide preliminary phenotypic insight into this unusual, but large, group of bacteria. Additionally, our method could be applied to other classes of uncultivated bacteria, many of which will be gradually added to the collection of genomes reconstructed from metagenomic sequencing of communities.

Another promising application of our approach is evaluating draft models during the metabolic network reconstruction process. In particular, in building new draft stoichiometric models, the producibility metric, which displays nuanced variability across taxa, could be used as an initial estimate of the biomass composition, to be compared to the reference biomass compositions currently used in most reconstructions (*Lakshmanan et al., 2019*). More generally, our approach fits into an emerging class of metabolic reconstruction and analysis methods that address uncertainty by statistically sampling ensembles (of environments, as done here; fluxes, as studied extensively [*Schellenberger and Palsson, 2009*]; or network reconstructions, as recently implemented [*Biggs and Papin, 2017*; *Machado et al., 2018*]). We envisage that the metabolic insight gained from the application of these methods will continue to help bridge the gap between top down studies and a mechanistic understanding of microbial community metabolism and dynamics.

## Materials and methods

### Method implementation

The framework for implementing our method was developed as several different modular functions that interact in a nested manner to run our analysis. The functions are written in MATLAB and interface with the COBRA toolbox (*Schellenberger et al., 2011*; *Heirendt et al., 2019*). The code is built around the COBRA toolbox commands *changeObjective* and *optimizeCbModel*. Thus, running our code requires installation of the COBRA toolbox. Additionally, the nonlinear fitting function utilizes the MATLAB function *lsqnonlin* for nonlinear least squared fitting. Additional functions were developed to implement our probabilistic framework and run our analysis method. We describe here each modular function, providing details on the computations performed. The full code for implementing our method, with examples for running the code, is available online at https://github.com/segrelab/biosynthetic_network_robustness (*Bernstein, 2019*).

### Algorithm functions

*feas* – This function determines if the production of a given target metabolite set is feasible given the metabolic network model with specified constraints. Flux balance analysis was used to determine the feasibility of production (*Orth et al., 2010a*). Flux balance analysis was chosen over the alternative network expansion algorithm due to its treatment of cofactor metabolites (*Kruse and Ebenhöh, 2008*). In network expansion, cofactors must be added to the network to 'bootstrap' metabolism, whereas in flux balance analysis any reaction utilizing a cofactor can proceed given that the cofactor can be recycled by a different reaction, which is a less restrictive constraint on the metabolic network flux. Furthermore, our implementation allows for inequality or equality mass balance constraints. Traditional flux balance imposes an equality mass balance which is often referred to as a steady state constraint. This constraint restricts the rate of change of all metabolite concentrations to be equal to 0. We provide the option of implementing inequality mass balance, which constrains the rate of change of metabolite concentrations to be greater than or equal to 0. In practice, inequality mass balance is implemented by adding unbounded exporting exchange reactions and calculating steady state solutions. We have implemented inequality mass balance for all of our calculations due to the

fact that we are analyzing local properties of the metabolic network (the production of a single metabolite) and do not want the network to be constrained by the global requirement to achieve steady state. During the production of a particular metabolite, the metabolic network is thus free to produce byproducts that are used elsewhere or secreted. To determine production feasibility, the export of a particular target metabolite is set to the objective function and maximized. If the maximal flux is greater than a hard-coded threshold (>0.001), then the target metabolite is considered to be feasibly produced. This function uses the COBRA commands *changeObjective* and *optimizeCbModel* to set and maximize the appropriate objective function. Mathematically, flux balance analysis is implemented as a linear programming problem with the following definition:

$$\textbf{\textit{maximize}}: C^T v$$

$$\textbf{\textit{subject to}}: Sv = 0 \;(\textit{equality mass balance}); or \; Sv \geq 0 \;(\textit{inequality mass balance})$$

$$\textbf{\textit{and subject to}}: lb \leq v \leq ub$$

Where: $C^T$ is the transpose of a column vector indicating which reactions are to be maximized. In this case, this specifies the exporting exchange reactions corresponding to the target metabolites. $v$ is a column vector of metabolic reaction fluxes. $S$ is the stoichiometric matrix describing the reactions present in the metabolic network (a metabolites by reactions size matrix). Each element in the matrix is the stoichiometry of a particular metabolite associated with a particular reaction. Negative values indicate that a metabolite is a reactant of that reaction being consumed, while positive values indicate that a metabolite is a product of that reaction being produced. $lb$ and $ub$ are the lower and upper bounds of all reactions, which define reaction reversibility or are set to -1000 and 1000 respectively when unbounded. Additional information on flux balance analysis can be found in this publication describing its implementation in detail (*Orth et al., 2010a*).

*rand_add* – This function is designed to give a random sample of input metabolites to be added based on the Bernoulli parameter for each input metabolite. This function uses the MATLAB *rand* function to choose a random number between 0 and 1 for each input metabolite. If this number is less than the Bernoulli parameter for that input metabolite, then the metabolite is added.

*prob* – This function utilizes *rand_add* and *feas* to determine the probability of producing the target metabolite given the input metabolite Bernoulli parameters, the metabolic network structure, and the specified constraints. A chosen number of random samples of input metabolites are generated by repeatedly running the *rand_add* function. The probability of producing the target metabolite is determined as the number of feasible trials divided by the total number of samples. The default number of samples used for the bulk of the analysis in this work was 50.

*calc_PM_fit_nonlin* – This function calculates the producibility metric (PM) for a specified metabolic network model and metabolite using an efficient nonlinear fitting technique. The nonlinear fitting algorithm estimates the PM by randomly sampling points on the producibility curve that fall near PM. The algorithm starts by sampling a point in the middle of the producibility curve ($P_{in}$ = 0.5) and then using the MATLAB function *lsqnonlin* to fit a sigmoidal curve to the sampled points of the producibility curve. The fit sigmoidal curve is then used to estimate a value for the PM. Next, a new sample point is obtained which is offset from the estimated PM value with some noise introduced with the specified noise parameter. In this way the algorithm converges on the PM value and samples points around PM, thus increasing the accuracy of its estimate with each iteration. The estimate converges when a specified n estimates of the PM value are all within a specified threshold. The default parameters associated with this function, used for the bulk of our analysis, were: noise = 0.3, n = 7, thresh = 0.01. The parameters chosen were selected by hand to provide reasonable performance.

*prep_mod* – This function is used to prepare the metabolic network model for analysis with our method. The input for this function is a COBRA model, which is saved as a MATLAB structure variable. This code has been developed and optimized to work with KBase generated metabolic networks and is not guaranteed to work with networks from other sources that have different naming conventions. The first modification to the networks is to find and turn off all exchange and maintenance reactions to standardize the network models. Second, the extracellular and intracellular metabolites are identified based on naming conventions and output from the function. Third,

exchange reactions are added for each metabolite (producing 1 unit of that metabolite), and a vector indicating the mapping from metabolites to these exchange reactions is output from the function. This vector is used by our method to control the presence and absence of input metabolites in the network model as well as to adjust the inequality mass balance constraints. The final output is a new network model which has been standardized for our method and in which the presence and absence of metabolites can be easily manipulated.

*find_PM_mods_mets* – This function is designed to facilitate the parallelization of the PM calculation. The function takes as inputs a directory of metabolic network models, a directory to store results, a list of target metabolite names, the index of the current network model and metabolite being analyzed and all of the specifications necessary for running *calc_PM_fit_nonlin*. The metabolite and model being analyzed can be changed dynamically to allow for parallelization. In addition to these inputs, this function has several inputs that allow for standard modifications to the PM calculation procedure. It allows for certain metabolites to be fixed on or off. It allows for several choices of metabolites to be added during the PM calculation process, including adding all intracellular or extracellular metabolites and including the target metabolite or not. It also allows for specification of the inequality mass balance constraint as either all metabolites set to inequality mass balance or all metabolites set to equality mass balance. Furthermore, it has a parameter for the number of runs to calculate the PM to obtain statistics regarding the variability of *calc_PM_fit_nonlin*. For the analysis done in this work: calculation of PM for single metabolites was done by adding all intracellular metabolites (excluding targets), the mass balance constraint was set to use inequality constraints for all metabolites, the number of runs was set to 10.

## Parallelization

We used the Boston University shared computing cluster to run our analysis for a large number of metabolic networks and metabolites. The calculation of the PM for each individual network model and metabolite can be run in parallel, vastly increasing the number of possible computations. The average runtime for computing the PM for an individual network and metabolite for 10 repeated runs was ~9 min and the maximum run time was ~45 min, given the default parameters used in this study: a = 0, s = 1, samp = 50, noise = 0.3, n = 7, thresh = 0.01, runs = 10. We note that these parameters were chosen by hand to provide adequate performance for our algorithm, and future implementations could possibly alter these parameters to provide improved run-time and/or accuracy.

## Using the *E. coli* core metabolic network to demonstrate features of metabolite producibility

Our analysis method was initially demonstrated on the *E. coli* core metabolic network. We used the network provided by the BiGG database (*King et al., 2016*). We calculated the PM value for each intracellular metabolite. The input metabolites for our PM calculations were assigned as all intracellular metabolites in the *E. coli* core metabolic network. This was the most naïve assumption we could use for assigning input metabolites, and was consistently used throughout the majority of our analyses. Additionally, using intracellular metabolites as input metabolites avoids errors that could arise from poorly annotated transporters in draft metabolic network reconstructions. Calculations were performed using the Boston University shared computing cluster to parallelize runs across networks and metabolites and improve computation time. The results of our simulation were visualized using the Cytoscape network visualization software (*Shannon et al., 2003*). The entire *E. coli* core metabolic network is shown, excluding the biomass reaction for clarity.

## Producibility of metabolites differs from pathway completeness and captures minimal precursor set structure

We analyzed the PM for the histidine biosynthetic pathway with auxotrophic metabolic networks generated by knocking out different reactions along the pathway in the *E. coli* iJO1366 metabolic network. The PM was calculated for all essential biomass components using default parameters. The PM for all biomass components, excluding histidine, was unchanged and the PM for histidine was reported. The PM for histidine was seen to match the theoretical values based on our combinatorial theory. The theoretical values were calculated using the formula in *Figure 1—figure supplement*

*2 C*, where *n* corresponds to the number of main intermediate metabolites that remain connected to the end product (L-histidine). For our analysis of the histidine pathway across all 456 oral microbiome metabolic networks, the histidine pathway sum was the total number of reactions in the histidine pathway and the pathway length was the total number of reactions starting at histidine and counting until the first missing reaction. Both values were compared to the PM values of each organism for histidine by Spearman's rank correlation.

## Producibility analysis shows improved tolerance to missing reactions compared to flux balance analysis

We demonstrate the performance of our method on a network with missing reactions on perturbed *E. coli* iJO1366 metabolic network by randomly removing reactions and observing the results of flux balance analysis (FBA) and our producibility metric (PM). The iJO1366 metabolic network consists of 2583 reactions, and was perturbed by removing *n* random reactions (*n* = 4, 16, 64, 256, 1024). The biomass reactions (full and core) and the maintenance reaction were not candidates to be randomly removed. For each value of *n*, 50 different randomly perturbed metabolic networks were generated (a total of 250 metabolic networks). For each of these networks, the core biomass reaction flux was optimized using FBA in a complete medium and a minimal glucose medium and the biomass flux was recorded. Next, the PM for all core biomass metabolites, excluding those with consistent PM of 0 (ex. metal ions), were calculated using our method. The accuracy of each method was calculated using a quantitative difference measure and a biomass production measure. For FBA the quantitative difference measure was calculated as the absolute value of the difference between the original bio-mass flux and the perturbed biomass flux. For the PM the quantitative difference measure was calculated as the sum of the absolute value of the differences between the PM value of each metabolite in the original model and the perturbed models. Note that for both FBA and the PM this measure is equivalent to the L1 norm of the difference between the original network metric and the perturbed network metric. The accuracy was measured as one minus the normalized difference measure. Means and standard errors across the 50 different randomly perturbed metabolic networks were calculated and reported. The biomass production measure was calculated in FBA as the fraction of perturbed metabolic networks that could produce biomass flux greater than 1% of the original optimal biomass flux. The biomass production measure was calculated for the PM as the fraction of perturbed metabolic networks that had PM for all biomass components analyzed above a specified threshold. Thresholds of 0.1 and 0.6 were analyzed and reported.

## Metabolite producibility points to putative metabolic mechanisms for *E. coli* auxotroph co-cultures

We analyzed experimental data from *E. coli* auxotrophs using our PM metric. Experimental data were taken from the supplementary growth data of *Wintermute and Silver (2010)*. The growth data used were the mean of the day 4 replicates 1 and 2. *E. coli* auxotrophs were modeled using the iJO1366 metabolic network. All reactions related to a gene, including those involving isoenzymes, were knocked out from the model by setting the upper and lower bound of the reaction to zero. Iso-enzyme related reactions were included based on prior evidence that this improves performance of metabolic modeling of auxotrophs (*Jacobs et al., 2017*). The PM was calculated for all biomass components for each auxotroph metabolic network using default parameters consistent with other PM calculations in this study. The PM distance between auxotrophs was calculated as the L1 norm of the difference between two auxotrophs PM vectors. Additional details on PM distance can be found in methods section *Metabolic similarity correlates with microbial co-occurrence in the human oral microbiome*. The correlation between PM distance and experimentally measured growth matrices was assessed using a Mantel permutation test with 10,000 permutations, and calculating the correlation of the upper triangle of the matrices. Additional details on the Mantel permutation test can be found in the methods section *Metabolic similarity correlates with microbial co-occurrence in the human oral microbiome*.

## Reconstruction of human oral microbiome metabolic networks

A set of 456 draft metabolic networks were reconstructed for oral microbiome strains. Strains were chosen to match the sequences chosen for dynamic annotation on HOMD which cover at least one

strain for each sequenced species and repeated strains for sequences of particular interest for the human oral microbiome. Several strains were additionally selected due to our interest in fastidious and uncultivated organisms. These included eight uncultivated or recently co-cultivated strains. When considering the taxa TM7 and *Tannerella* sp. oral taxon 286, we chose to include the most recent genome sequences from co-cultivation experiments, although there are several additional single-cell and metagenome assembled sequences available for *Tannerella* sp. oral taxon 286 and TM7 (*Kantor et al., 2013*; *Marcy et al., 2007*; *Beall et al., 2014*; *Albertsen et al., 2013*; *Podar et al., 2007*). The host strains *Actinomyces odontolyticus* XH001, *Pseudopropionibacterium propionicum* F0700, and *Pseudopropionibacterium propionicum* F0230a were included due to their support of TM7 organisms. All genomes were either found in the KBase central data repository or manually annotated with RAST and uploaded to KBase (*Arkin et al., 2018*; *Overbeek et al., 2014*; *Aziz et al., 2008*; *Brettin et al., 2015*). Strains that were dynamically annotated on HOMD but could not be found on KBase, were not of interest due to uncultivability, and already had a representative strain from their matching species were not included in our set of strains. Several naming discrepancies existed between KBase and HOMD, which are highlighted in the KBase download notes column of *Supplementary file 4*. All metabolic networks were reconstructed using a KBase narrative containing all of the genomes and metabolic networks from this work, which is available to be copied, viewed, edited, or shared at https://narrative.kbase.us/narrative/ws.27853.obj.935. Metabolic networks were reconstructed for each strain with automatic assignment of Gram-stain, and without gap-filling. Metabolic network reconstructions were then downloaded from KBase as SBML files and converted to COBRA. mat files using the COBRA command *readCbModel*. Metadata related to all organisms and metabolic networks are available in *Supplementary file 4*.

## Large-scale patterns in biosynthetic capabilities identified across the human oral microbiome

We investigated the large-scale biosynthetic properties of the human oral microbiome by analyzing reconstructed metabolic networks for 456 different oral microbiome strains. For each metabolic network we calculated the PM value for 88 individual biomass components (40,128 total PM calculations). The biomass components were chosen to be the union of the set of default KBase Gram-positive and Gram-negative biomass compositions (see *Supplementary file 5* for details). The metabolites sulfate and phosphate were not included, while the metabolite $H_2O$ was included as a positive control. The calculations were parallelized across metabolic networks and metabolites using the Boston University shared computing cluster to improve computation time. The PM values were stored as a matrix of organisms by metabolites PM values. This matrix was analyzed using hierarchical clustering based on average differences between groups. The matrix was clustered and visualized using the R package *pheatmap*.

For the comparison of average PM values and genome size, genome size was taken from KBase and added to *Supplementary file 4*. We used regression modeling to identify the broad relationship between genome size, taxonomy, and the average PM value. We fit PM values to linear and quadratic models of log genome size:

$$\text{Linear} : \text{average}(\text{PM}) = c1 + c2 * \log(\text{genome size})$$

$$\text{Quadratic} : \text{average}(\text{PM}) = c1 + c2 * \log(\text{genome size}) + c3 * \log(\text{genome size})^2$$

Nominal taxonomic parameters were added to these models to determine if they could improve the models prediction of PM values. Gram-stain, and taxonomic labels from phylum to genus were used as nominal taxonomic parameters. For each taxonomic level, each label was added as an additional nominal parameter, for example: adding the predictor of phylum meant adding 12 independent variables, one for each different phylum. Gram-stain was assigned based on KBase default assignments. Taxonomic labels were assigned based on human oral microbiome database taxonomy annotations. Regression models were developed using the MATLAB command *fitlm*. The AIC and BIC were calculated to assess model improvement upon subsequent addition of taxonomic parameters using the MATLAB command *aicbic*.

## Taxonomic trends capture biosynthetic patterns across human oral microbiome organisms

We investigated specific trends in metabolite PM values related to taxonomy by analyzing the clustered matrix of PM values. Additionally, a regression model was used to provide quantitative insight. The base regression model was a quadratic model using the log of genome size as the predictor of the specific PM value for a certain metabolite across all organisms:

$$PM(metabolite) = c1 + c2*\log(genome\ size) + c3*\log(genome\ size)^2$$

Nominal taxonomic parameters were then added one at a time. Taxonomic parameters of Gram-stain (+ or -), phylum (belonging to 1 of 12 phyla or not) and class (belonging to 1 of 22 classes or not) were used. We calculated the log likelihood ratio by taking difference between the log likelihood of the base quadratic model of genome size and the model including a specific taxonomic parameter. We identified highly significant relationships using an alpha value of $10^{-6}$ and Bonferroni correction for multiple hypothesis testing.

## Organic acid production predicted for human oral microbiome organisms

Organic acid production was assessed by calculating the PM for nine different organic acids for each human oral microbiome organism. The organic acids analyzed were: acetate, formate, L-lactate, succinate, propionate, D-lactate, butyrate, isobutyrate, and isovalerate. The organic acids were chosen by searching through the literature for those that were found to be relevant for oral health (*Jorth et al., 2014*; *Takahashi, 2015*). While this is certainly not an exhaustive list of organic acids, it demonstrates the applicability of our method to non-biomass metabolites. The PM was calculated using default parameters consistent with other calculations in this study. The organic acids chosen where included in *Supplementary file 5*, and the PM results were included in *Supplementary file 6*.

## Metabolite producibility in a protein vs. carbohydrate-enriched environment

The production of metabolites in variable environments was implemented in our method by re-calculating the PM for all metabolites in a protein-enriched and carbohydrate-enriched environment for two organisms: a saccharolytic organism (*Streptococcus mutans* UA 159) and a proteolytic organism (*Porphyromonas gingivalis* W83). These organisms were chosen because they are known to be associated with oral diseases involving either saccharolytic or proteolytic activity, namely dental carries and periodontitis (*Wade, 2013*; *Takahashi, 2015*). The protein-rich environment was simulated by fixing all 20 amino acids to always be present ($P_{in}$ = 1), by adding them to the *fixon* parameter, and then adding all other metabolites randomly. As in the other PM calculations, the target metabolite is never added or fixed to be on. The carbohydrate-enriched environment was simulated in a similar manner by fixing D-glucose to always be present.

## Metabolic similarity correlates with microbial co-occurrence in the human oral microbiome

Co-occurrence data were collected from supplementary Dataset_S1 of *Friedman and Alm (2012)*. Seven different oral sites were analyzed and co-occurrence calculated with SparCC and Pearson's correlation were analyzed. Various pairwise metabolic metrics where compared to the patterns of microbial co-occurrence, and significant correlations were found using a Mantel permutation test with 10,000 permutations. These pairwise metabolic metrics were likewise compared to each other using a Mantel permutation test with 1000 permutations. Additional details on metrics used and Mantel test are described below. To compare the pairwise metabolic metrics with microbial co-occurrence, we collapsed all interaction metrics to the genus level by averaging scores across species in the same genus such that we could match predictions from our method with co-occurrence based on 16S rRNA sequencing, which was mapped at best to the genus level.

Calculation of various metrics for correlation analysis:

*PM distance* – Calculated as the L1 norm of the difference between the PM vectors of any two organisms. The L1 norm is calculated as the sum of the absolute values of the differences of all of the elements of the vector.

$$PM\ distance_{A,B} = norm_{L1}(PM^A - PM^B) = \sum_{i=1}^{n}|PM_i^A - PM_i^B|$$

*PM complementarity* – Calculated as the complementarity between two organisms, quantifying the amount by which one organism can supplement the PM of another organism. For organism A supplementing B, the metric is calculated by summing over i = 1 to N metabolites.

$$PM\ complementarity_{A \to B} = \frac{\sum_{i=1}^{N}(\max[PM_i^A, PM_i^B] - PM_i^B)}{\sum_{i=1}^{N}(PM_i^A)}$$

*Seed distance* – Calculated as the L1 norm of the difference between the seed vectors of any two organisms. The seed vectors were calculated using the NetSeed method (*Borenstein et al., 2008*; *Carr and Borenstein, 2012*). The code used to calculate the seeds was taken from http://elbo.gs. washington.edu/software_netcooperate.html. The *minComponentSize* parameter was set to 0 and the *onlyGiant* component parameter was set to false.

*Seed competition* – Calculated following the formula used in the NetCmpt method (*Kreimer et al., 2012*; *Levy and Borenstein, 2013*). The competition from A to B is the fraction of seeds of organism A that are also seeds of organism B. The threshold for seed vs. non-seed was a seed score of greater than 0.

*Seed complementarity* – Calculated following the formula used in the NetCooperate method (*Levy et al., 2015*; *Levy and Borenstein, 2013*). The complementarity from A to B is the fraction of seeds of organism A that are in the metabolic network of organism B but not seeds of organism B. This metric was calculated using code from http://elbo.gs.washington.edu/software_netcooperate. html.

*Reaction distance* – Calculated as the L1 norm of the difference between the reaction vectors of any two organisms. Reaction vectors were vectors of 0's and 1's indicating which metabolic reactions where present in the draft metabolic network of each organism.

*Reaction Jaccard* – Calculated as the Jaccard distance between the reaction vectors of any two organisms. The Jaccard distance is calculated as one minus the intersect of the vectors divided by the union of the vectors. In other words, it is one minus the fraction of shared metabolic reactions.

*Mantel test* – A Mantel test was used to assess correlation between matrices as done in *Levy and Borenstein (2013)*. The Spearman's rank correlation was calculated for all elements of the two matrices (excluding the diagonal). Then the first matrix was permuted 10,000 times, and the number of times the correlation was stronger than the original correlation was recorded. The p-value was calculated using the formula below, where *n* is the number of times the permuted correlation was stronger (absolute value of the correlation coefficient ρ was larger) than the original and *N* is the number of permutations.

$$Mantel\ P\ value = \frac{n+1}{N+1}$$

Partial Mantel tests were calculated in a similar manner, but using partial correlations between the first and second matrix while controlling for a third matrix. For the partial correlation permutations, only the first matrix is permuted and the partial correlation is recalculated.

## Biosynthetic properties predicted in a cluster of fastidious human oral microbiome organisms

A subset of fastidious organisms identified from the larger clustered matrix of all oral microbiome organisms PM values were re-clustered based on average distances and analyzed further. Additionally, three previously uncultivated TM7 organisms (TM7x, AC001, and PM004) and several host strains for the uncultivated TM7 (*Actinomyces odontolyticus* XH001, *Pseudopropionibacterium propionicum* F0700, and *Pseudopropionibacterium propionicum* F0230a) were re-clustered and analyzed. Metabolites were ranked and analyzed based on the difference between the average PM value of separate groups. Three different rankings were used throughout this analysis: (1) average fastidious cluster organisms PM subtracted from average oral microbiome organisms PM, (2) average *Mycoplasma* PM subtracted from average TM7 PM (3), average TM7 host PM subtracted from

TM7 PM. Correlations between amino acid biosynthetic cost and difference in PM were calculated using Spearman's rank correlation between the amino acid cost (*Akashi and Gojobori, 2002*) and the difference in average PM between average and fastidious organisms, or between hosts and TM7, using the MATLAB command *corr*.

## Acknowledgements

We would like to acknowledge our collaborators at the Forsyth institute for providing valuable knowledge on the human oral microbiome and uncultivated microorganisms. In particular, we would like to acknowledge Andrew J Collins and Pallavi P Murugkar for providing genome information on their TM7 strains AC001 and PM004 in coculture with *Pseudopropionibacterium propionicum* ahead of their work being accepted for publication. We would also like to acknowledge all of the members of the Segrè lab and Daniel Sher of Haifa University for helpful discussions and comments on this work. Research reported in this publication was supported by The National Institute of Dental and Craniofacial Research of the National Institutes of Health under award numbers R37DE016937, R01DE024468, by National Institutes of Health grants R01GM121950 and Sub_P30DK036836_P and F, by the Defense Advanced Research Projects Agency (Purchase Request No. HR0011515303, Contract No. HR0011-15-C-0091), the US Department of Energy (DE-SC0012627), the Boston University Interdisciplinary Biomedical Research Office, and by the Boston University training program in quantitative biology and physiology under Ruth L Kirschstein National Research Service Award T32GM008764 from the National Institute of General Medical Sciences. The content is solely the responsibility of the authors and does not necessarily represent the official views of the granting agencies.

## Additional information

### Funding

| Funder | Grant reference number | Author |
|---|---|---|
| National Institute of Dental and Craniofacial Research | R37DE016937 | Floyd E Dewhirst |
| National Institute of General Medical Sciences | R01GM121950 | Daniel Segrè |
| Defense Advanced Research Projects Agency | HR0011-15-C-0091 | Daniel Segrè |
| Biological and Environmental Research | DE-SC0012627 | Daniel Segrè |
| National Institute of Dental and Craniofacial Research | R01DE024468 | Floyd E Dewhirst Daniel Segrè |
| National Institute of General Medical Sciences | T32GM008764 | David B Bernstein |
| National Science Foundation | 1457695 | Daniel Segrè |
| National Science Foundation | NSFOCE-BSF 1635070 | Daniel Segrè |
| Human Frontier Science Program | RGP0020/2016 | Daniel Segrè |

The funders had no role in study design, data collection and interpretation, or the decision to submit the work for publication.

### Author contributions

David B Bernstein, Conceptualization, Data curation, Software, Formal analysis, Investigation, Visualization, Methodology, Writing—original draft; Floyd E Dewhirst, Conceptualization, Supervision, Funding acquisition, Investigation, Writing—review and editing; Daniel Segrè, Conceptualization, Supervision, Funding acquisition, Investigation, Writing—original draft, Project administration

## Author ORCIDs

David B Bernstein (ID) https://orcid.org/0000-0001-6091-4021
Daniel Segrè (ID) https://orcid.org/0000-0003-4859-1914

## Decision letter and Author response

Decision letter https://doi.org/10.7554/eLife.39733.030
Author response https://doi.org/10.7554/eLife.39733.031

# Additional files

## Supplementary files

• Supplementary file 1. Visualization of TM7 metabolic networks and producibility metric: TM7x HMT952. Cytoscape files are provided for visualizing the metabolic networks of the TM7 organisms analyzed in this work. The producibility metric (PM) for all intracellular metbolites is shown for each network. Networks are represented as bipartite networks with reactions as squares and metabolties as circles. Black borders are used to indicate irreversible reactions, reversible reactions have no border. Intracellular metabolite color indicates calculated PM (blue = low, red = high).
DOI: https://doi.org/10.7554/eLife.39733.021

• Supplementary file 2. Visualization of TM7 metabolic networks and producibility metric: AC001 HMT488. Cytoscape files are provided for visualizing the metabolic networks of the TM7 organisms analyzed in this work. The producibility metric (PM) for all intracellular metbolites is shown for each network. Networks are represented as bipartite networks with reactions as squares and metabolties as circles. Black borders are used to indicate irreversible reactions, reversible reactions have no border. Intracellular metabolite color indicates calculated PM (blue = low, red = high).
DOI: https://doi.org/10.7554/eLife.39733.022

• Supplementary file 3. Visualization of TM7 metabolic networks and producibility metric: PM004 HMT955. Cytoscape files are provided for visualizing the metabolic networks of the TM7 organisms analyzed in this work. The producibility metric (PM) for all intracellular metbolites is shown for each network. Networks are represented as bipartite networks with reactions as squares and metabolties as circles. Black borders are used to indicate irreversible reactions, reversible reactions have no border. Intracellular metabolite color indicates calculated PM (blue = low, red = high).
DOI: https://doi.org/10.7554/eLife.39733.023

• Supplementary file 4. Oral microbiome organisms' metadata. Metadata related to the oral microbiome organisms analyzed in this work is provided.
DOI: https://doi.org/10.7554/eLife.39733.024

• Supplementary file 5. Metabolites' metadata. Metadata related to the metabolites analyzed in this work is provided.
DOI: https://doi.org/10.7554/eLife.39733.025

• Supplementary file 6. Producibility metric matrix. The producibility metric (PM) for each organism (shown in *Figure 4*) is provided.
DOI: https://doi.org/10.7554/eLife.39733.026

• Supplementary file 7. Correlation of inter-organism metabolic metrics with co-occurrence. Correlation data from the analysis presented in Results section '*Metabolic similarity correlates with microbial co-occurrence in the human oral microbiome*' is provided. Correlations between inter-organism metabolic metrics and microbial co-occurrences from *Friedman and Alm (2012)* were calculated using a Mantel test with 10,000 permutations.
DOI: https://doi.org/10.7554/eLife.39733.027

• Transparent reporting form
DOI: https://doi.org/10.7554/eLife.39733.028

## Data availability

All scripts and metabolic network data used for generating the manuscript results are available on GitHub (https://github.com/segrelab/biosynthetic_network_robustness) (f82f1e0; copy archived at

https://github.com/elifesciences-publications/biosynthetic_network_robustness). All genomes used to derive the metabolic networks are available from the Human Oral Microbiome Database (http://www.homd.org/), except for three strains whose genomes are available on NCBI GenBank, with the following accession numbers: Saccharibacteria (TM7) bacterium HMT-488 strain AC001: NCBI CP040003, Saccharibacteria (TM7) bacterium HMT-955 strain PM004: NCBI CP040008, Pseudopropionibacterium propionicum HMT-439 strain F0700: NCBI CP040007. The data shown in the figures are also available in the form of supplementary tables included in the article.

The following datasets were generated:

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
