## [Decision Letter]

[Editors’ note: the authors were asked to provide a plan for revisions before the editors issued a final decision. What follows is the editors’ letter requesting such plan.]

Thank you for sending your article entitled "Quantifying biosynthetic network robustness across the human oral microbiome" for peer review at *eLife*. Your article is being evaluated by three peer reviewers, one of whom is a member of our Board of Reviewing Editors, and the evaluation is being overseen by a Reviewing Editor and Naama Barkai as the Senior Editor.

Given the list of essential revisions, including new experiments, the editors and reviewers invite you to respond with an action plan and timetable for the completion of the additional work. We plan to share your responses with the reviewers and then issue a binding recommendation.

The most important issue you will need to address is the biological relevance of the proposed method. You can either show that the method is in agreement with previously published data (where a lot of direct evidence exist) in a manner that is better/more insightful than previous methods; or provide new experimental data to substantiate your claims.

*Reviewer #1:*

"Quantifying biosynthetic network robustness across the human oral microbiome" from Segre lab was motivated by a desire to overcome the gap-filling limitation of standard flux balance analysis and the need to make assumptions on environments in metabolic network topology analysis. They quantified "biosynthetic network robustness", essentially how the probability of a product being made (*P_out_*) scales with the probability of the presence of input metabolites (*P_in_*). This robustness can be summarized as PM – the higher PM, the more likely a metabolite can be synthesized. If one is worried about parameter overfitting (and one should worry about such problems in genome scale models), this seems like a good thing to try. Applying the metric PM to *E. coli* central metabolism, they found that PM is not correlated with connectivity. Some metabolites have high connectivity but low PM, and some low connectivity and high PM. They then quantified PM of 88 biomass metabolites for 456 oral microbes, and found that fastidious (difficult to culture) microbes have low PM values for several metabolites, consistent with them being fastidious. Genome size and taxonomy both contained information about PM. They found that cell wall metabolites have variable PM. They also found that a number of amino acids have high PM in *Proteobacteria* and low PM in *Bacteroidetes*. They also analyzed metabolic complementarity of several TM7 strains (fastidious microbes) and their symbiotic bacteria strains. The paper is of bioinformatics nature, and is in general clearly written.

1) What is the difference between PM and% completion of biosynthetic pathway?

2) It's also not entirely clear to what extent their method improves upon the state of the art. In their Introduction, they say that two downsides that limit FBA (but not their method) are assumptions on environmental conditions and reconciling with stoichiometry-based constraints. I don't really understand their second limitation, but they sidestep the first limitation by assuming a constant *P_in_* among all inputs. This seems similar to assuming constant concentration among all inputs for FBA, which is never shown.

*Reviewer #2:*

The manuscript by Bernstein et al. presents a computational method to estimate metabolites that are likely to be exchanged in a microbial community. The essence of the novelty of the method, especially in comparison with the previous methods, is that it provides a probabilistic view accounting for multiple possible variations in substrate uptake combinations (e.g. different combinations of C/N sources etc.). The method is illustrated with the analysis of oral microbial community and the results suggest potential metabolic dependencies in the community. I find the method to be elegant but yet only of theoretical interest.

1) The method assumes a uniform distribution of possible substrate uptake combinations – which is not necessarily biologically observed. There are hierarchies – e.g. preference of glucose over other C-sources in most organisms.

2) The finding that metabolic dependencies may exist in oral microbiome is not that surprising per se given that a large body of literature (including papers also from the Segre lab) already suggest that metabolic dependencies are likely widespread in microbial communities. Thus, it is not clear whether only the proposed method would be able to identify these.

3) Lack of experimental validation makes it difficult to assess whether the identified exchanges are happening in the system. At the least, a detailed analysis of systems with known metabolite exchanges (and how the proposed method there differs from the previous ones) should be provided.

In summary, I do find the proposed method elegant and appealing but miss substantiation of the biological relevance.

*Reviewer #3:*

This manuscript introduces a new robustness metric that attempts to quantify the probability that an organism can produce a given metabolite directly from the genome. This probability is calculated as an average over possible environments defined by the presence or absence of the other metabolites in the network with production determined through the application of an FBA model. I found the metric itself intuitively appealing and easy to understand. In addition, the manuscript was clear and well written. The application to the oral microbiome and the novel CPR organisms in particular was also well motivated and interesting.

Given the rate of generation of microbial genomes it would be very useful to have a better method to even approximately predict metabolic products of organisms. However, what the study lacks in my opinion, is quantitative evidence that the metric actually does translate into a prediction that the organism produces a given metabolite. They provide some anecdotal examples where it appears to work but without a more thorough analysis of what the metric means then it will only be useful in a qualitative sense i.e. a higher PM indicating that one organism is more likely to produce a metabolite than another. Metabolic modelling is admittedly not my field, but datasets capable of testing the metric, must exist. A possible example might be this collection of *E. coli* strains and phenotypes defined in terms of growth conditions (Galardini et al., 2017, *eLife*). If such datasets are not available then I feel the authors need to be more circumspect in their conclusions and concede that the metric may say something about the production of metabolites but without any certainty about how a given PM value translates into an actual probability of production.

Other than the above caveat, I thought this was a good study but it was sparse on the practical details of the methodology. The GitHub repository consists of a selection of Matlab scripts without a clear description of how to apply them. What would be really useful is a complete walkthrough of the methodology from genome to PM values including for example ggkbase commands used to build the models. This is necessary if this work is to be properly reproducible. One final comment was that some evaluation of the impact of sample number on the calculated PM values would be a worthwhile addition. The authors mention that 50 samples of input metabolite combinations were used but no analysis of the impact of this choice on the PM values was given.

[Editors’ note: formal revisions were requested, following approval of the authors’ plan of action.]

Thank you for submitting a revision plan for your article entitled "Quantifying biosynthetic network robustness across the human oral microbiome". The editors and reviewers have considered your plan and invite you to proceed with your revisions. Please also consider the following comments when preparing your revised manuscript.

The Reviewing Editor stated:

"I think that I know how authors can fix their problems from a theoretical aspect. They can start with a complete model as the "ground truth" model, but hide or mis-annotate various% links to mimic our incomplete knowledge of the system, and compare their method with existing methods. This way, they will always have a "ground truth" curve for any metric of interest to biologists. Of course, the task of assigning missing links will need to be repeated many times and in double-blind fashion to gain statistical power. I suspect that different methods might show tradeoff. For example, in their Figure 1, it is not clear whether the extra info provided by PM is useful to a biologist if all mutants are auxotrophic. Any tradeoff between methods is also valuable to know. Authors can also use this ground truth model to illustrate problems of gap filling etc."

Another reviewer said:

"I agree this is a very comprehensive plan and they do seem to have taken our comments to heart and developed a proposal to address them. I was pleased with their suggestions regarding an alternate dataset of *E. coli* phenotypes to test their metric and the improved GitHub repository. "

The last reviewer said:

"I agree that the proposal is comprehensive and I like the Reviewing Editor's simulation idea. This also reminds me that the authors' should comment on the ensembleFBA approach which does address the main criticisms of the gap-filling. Furthermore, I think that a clear focus on oral microbiome would benefit the study much more rather than claiming general validity that is very difficult to substantiate based on their proposed plan – after all, *E. coli* is one organism and by no means representative of natural diversity and community complexity. *E. coli* auxotroph complementation experiments are also unique in some sense as I haven't seen many other studies showing similar data in other organisms. So, overall, I think recommending to focus on oral microbiome and reducing the general claims would put the paper on more solid foundation I believe."

---

## [Author Response]

[Editors’ note: what follows is the authors’ plan to address the revisions.]

Thank you very much for evaluating our manuscript, and for your valuable and constructive comments. In the summary letter, the reviewers and reviewing editor have asked us to further investigate and clarify the novelty and biological relevance of our method, especially in relation to previously developed approaches. In particular, they have asked us to “either show that the method is in agreement with previously published data (where a lot of direct evidence exist) in a manner that is better/more insightful than previous methods; or provide new experimental data to substantiate [our] claims”.

We agree that a better description of the biological relevance of our method and additional analyses directly addressing how our method can provide novel insight relative to existing approaches will greatly strengthen our results and clarify the importance of our work. We propose several new analyses that will address the reviewers’ concerns. Based on revisiting the literature, and implementing some preliminary analyses (shown below), we believe that sufficient published data is available to clarify the biological relevance and distinct capabilities of our method.

The current document outlines our plan for addressing the reviewers’ comments, and is structured as follows:

In Part 1, we present an overview of how we will address each of the major issues raised across all reviewers. In Part 2, we have specifically addressed all reviewer’s comments in more detail. In Part 3, we have included preliminary and prospective figures that will be added to our manuscript to address the reviewer’s comments.

We obviously cannot guarantee that all comparisons with available data will confirm the validity of our approach. However, some preliminary analyses we have performed so far, displayed below in the form of tentative figures, support and clarify the relevance of our method. Would any of the prospective analyses we have suggested yield uninformative or negative results, we would still be committed to including them in a revised manuscript as an opportunity to highlight the limitations of our approach, and possible directions for future improvement.

Part 1 – Summary of how we address major issues

A) It is not clear how our method is different from and better/more insightful than previous methods (reviewer #1 comment 1, reviewer #1 comment 2, reviewer #2 comment 2).

We appreciate this criticism and see it as an opportunity to clarify the novelty of our method and point out where it may be better/more insightful than other similar methods. We will address this criticism by describing how our method differs from several different methods:

How our method differs from pathway completeness: We will first analyze the relationship between pathway completeness (the fraction of genes or reactions from a given pathway found in a metabolic network) and our Producibility Metric (PM) in depth for a single pathway. Figure 1 displays a preliminary analysis, showing how our metric provides a much more informative outcome than pathway completeness for the analysis of mutations in the *E. coli* L-histidine biosynthesis pathway. In particular, the PM provides information on the structure of the pathway and on how the proximity of the knockout to the final product affects its producibility. Thus, a high PM value would indicate that the target metabolite can be produced from multiple distinct sets of precursors. This analysis will be presented as a supplemental analysis in the Materials and methods section of our manuscript.

To show the validity of this argument beyond a simple linear pathway and perform a more comprehensive analysis, we will also compare the relationship between pathway completeness and our PM for additional metabolites with known biosynthetic pathways and interesting distributions of PM across our set of oral microbiome organisms. This analysis will be presented in the results section of our manuscript. See also response to reviewer #1 comment 1 for additional details on these analyses.

How our method differs from flux balance analysis (FBA): We will clarify how our method differs from classical FBA, in three major ways. First, in our approach we assess the producibility of individual biomass components, rather than optimizing the overall biomass production flux. This allows us to analyze incomplete networks that would be simply infeasible with FBA optimization of the biomass production flux. Second, we define environmental availability of precursors in a probabilistic way, and thus compute metabolite producibility based on random sampling. Third, we provide a novel measure of robustness (the Producibility Metric) that cannot be computed with standard FBA, which describes how well an organism can produce a metabolite across a statistical ensemble of possible environments.

In the revised manuscript, in addition to describing in detail these differences and their implications, we will add a new figure supplement to our method section (see proposed Figure 2 below as Author response image 2) showing how indeed our metric provides robust predictions of biomass producibility for metabolic networks whose gaps would prevent obtaining any useful information from regular FBA (which would give simply “infeasible” as an outcome). Additionally, to clarify the connection between our probabilistic definition of environments and the constraint-based definition of FBA, we will further compare our analysis with FBA for *E. coli* grown on minimal glucose and rich media (see also point B below). In the discussion of a revised manuscript we would also better explain that having to perform gap-filling in order to use FBA for any microbiome study is becoming an increasingly important and debated issue. While previous work has shown the utility of gapfilled models for predicting inter-species interactions (Zelezniak et al.,2015; Freilich et al., 2011), more recent papers have raised concerns about existing gap-filling pipelines (Karp et al., 2018, BMC Systems Biology). In general, we feel that exploring alternative strategies that can estimate metabolic properties of microbes and communities without gap-filling is going to be a crucial component of future genome-scale microbial modeling efforts.

How our method differs from NetCooperate: We will compare our method in more detail to what we believe to be the most closely related method, in terms of scope and motivation, namely the NetCooperate reverse ecology method. Similar to our method, this approach can be used to analyze metabolic interactions in microbial communities using draft metabolic networks and to provide large scale insight into microbial communities (Levy and Borenstein, 2013; Levy etal., 2015). To perform a comparison with this method, we will use our method to compute a complementarity score (which quantifies the potential for metabolic exchange between two organisms) and compare this metric to an analogous metric derived from NetCooperate. We will compare the two approaches for all pairs of oral microbiome organisms in our collection. A tentative new figure (proposed Figure 3 below as Author response image 3) shows preliminary results from such an analysis. This figure will be highlighted in a supplementary results section of our revised manuscript. One can see that the two metrics are significantly correlated. Interestingly, however, our metric provides a better capacity to discriminate interactions that involve fastidious/uncultivated organisms. In finalizing this analysis, we will further check whether the capacity of our method to resolve uncultivated organisms is related to the more specific predictions of exchanged metabolites that our method provides relative to NetCooperate, likely due fact that we use stoichiometric constraints for our calculations, as opposed to network adjacency. To clarify why our analysis of the full stoichiometric network is expected to yield more specific prediction than those obtained by analyzing a metabolite adjacency network we propose to add a new portion of text, possibly aided by a figure supplement in the Materials and methods section. More details on this are provided in the response to reviewer #1 comment 2.

B) The assumption of a constant *P_in_* throughout our analysis seems limiting (reviewer #1 comment 2, reviewer #2 comment 1).

We wish to start by acknowledging that, while in our manuscript we described in detail the mathematical definitions of *P_in_* (the probability that an environmental metabolite is available) and of PM (the Producibility Metric), we could have done a better job at providing a biological intuition for our definitions. We will update our manuscript to include an improved description of PM, *P_in_*, and their relationship to FBA, which we outline tentatively in what follows.

One way of briefly re-stating the interpretation of the Producibility Metric (PM) for a given metabolite is that it is an estimate of the fraction of distinct environmental compounds that one would have to add in the medium in order to expect, on average, a high chance (above 0.5) of being able to produce a given metabolite. We should clarify that the computation of PM is effectively the outcome of thousands of FBA calculations in which the production of a target metabolite is estimated under many sets of input metabolites randomly sampled as *P_in_* slides between 0 and 1. In this sense, in our current calculations, *P_in_* is uniform (i.e. assumed to be the same across all metabolites), but not constant (it changes between 0 and 1). Note also that while the PM of a given metabolite is the outcome of many FBA calculations, it is a number that could not easily be related to any individual FBA calculation. In summary, PM quantifies the overall propensity for an organism (with an incomplete metabolic network, and no knowledge about its typical environment) to be able to produce a given metabolite. Therefore, our method is particularly useful for assessing the biosynthetic capabilities of a large number of organisms without having to make any arbitrary assumption beyond the metabolic information present in the organisms’ genomes.

That being said, nothing in our algorithm prevents one from making alternative choices for the values of *P_in_*, which can be different for different metabolites. In fact, our algorithm was designed in a modular fashion such that it could accommodate such alternative implementations. An extreme, but illustrative, scenario is the one of *P_in_* = either 0 or 1 for all metabolites. We will show that this scenario would effectively be equivalent to estimating the producibility of biomass components with standard FBA, under a specific environmental condition, in which the available metabolites are the ones with *P_in_* = 1. We will further show how these extreme choices of *P_in_* differ from using intermediate *P_in_* values and PM by showing how the results of metabolite producibility in a minimal and rich medium are related to the PM. This analysis is described in further detail in response to reviewer #1 comment 2.

In order to address the specific suggestion to use non-uniform *P_in_* values to probe nutrient preferences across different organisms, we will test the ability of our approach to distinguish between saccharolytic vs. proteolytic organisms in the human oral microbiome. These two classes of organisms have very different metabolic strategies and important implications in oral health (Takahashi, 2015). For this analysis, we will use variable *P_in_* values to define a sugar rich and an amino acid rich environment. Then we will calculate the producibility metric for different biomass components for a set of saccharolytic and proteolytic organisms from our collection of oral microbiome organisms and assess our ability to distinguish their environmental preferences. This analysis will be added to the results section of our manuscript in a new section addressing variable *P_in_* values. For additional details see prospective Figure 4 legend below.

C) There is a lack of experimental validation about producible metabolites and exchange among microbial organisms (reviewer #2 comment 3, reviewer #3 comment 1).

In general, experiments that directly confirm the exchange of a specific metabolites between two microbes are quite challenging, let alone experiments that could uncover the entire possible network of metabolic exchanges in a microbial community. However, a few published examples contain enough information to enable a direct comparison between computational and empirical outcomes. We propose to analyze different sources of existing data in order to validate the predictions of our algorithm. Whenever possible, during these analyses, we will also include analysis based on the methods mentioned in part A such that we can compare the performance of our method relative to these alternative methods. The specific sets of case studies we plan to focus on are described below:

We will analyze organic acid exchange in the oral microbiome: We will test our method’s ability to predict the exchange of organic acids between oral microbes in two known examples. To do this we will extend our method to predict the producibility of metabolites beyond biomass components such as organic acids. The specific interactions we will focus on are the exchange of lactic acid, produced by *Streptococcus* and consumed by *Veillonella* species (Chalmers et al., 2008); and the exchange of succinate from *T. denticola* to *P. gingivalis* and isobutyric acid from *P. gingivalis* to *T. denticola* (Grenier, 1992, Infect. Immun.).Additionally, following the analysis of these specific experimentally validated systems, we will extend our analysis to all of our oral microbiome organisms to generate hypotheses regarding exchange of organic acids between previously undiscovered pairs of organisms. This analysis will be presented in the manuscript as a supplemental result that expands on our current matrix of biomass producibility. See the response to reviewer #2 comment 3 for further details.

We will analyze the exchange of amino acids in *E. coli* auxotrophs: As a slightly more complex example of a microbial community with metabolic exchange we propose to analyze a dataset on putative exchange among amino acid auxotrophs (Mee et al., 2014). We will calculate the producibility metric for all amino acids across pairs and triplets of *E. coli* auxotrophs that are known to support each other through syntrophic exchange of amino acids. We will determine if indeed the production of the necessary amino acids to support partner growth is predicted by our method. This analysis will be added to the manuscript method section as an additional validation. See prospective Figure 5 legend and the response to reviewer #2 comment 3 for further details.

While the above analyses are focused on specific interactions for which molecular mechanistic details can be addressed with our methods, it will be also interesting to assess the performance of our method on microbial co-occurrence data from the human oral microbiome. It is important to note that co-occurrences are not necessarily indicative of metabolic interactions, but they are often related to metabolic trends, as shown before (Levy and Borenstein, 2013). We will utilize two different sources of microbial cooccurrence data for this analysis. The first will be high-throughput sequencing data from the human microbiome project, which contains co-occurrence data for microbes associated with several different human oral microbiome sites (Huttenhower et al., 2012, Nature). The second source we will aim at utilizing is a set of CLASI-FISH experiments of dental plaque where physical interactions between different taxa have been imaged through fluorescence microscopy (Welch et al., 2016). This analysis will be added to the results section of our manuscript. See response to reviewer #2 comment 3 and prospective Figure 6 legend for further details.

An additional analysis that we will pursue, as inspired by reviewer #3 comment 1, is the assessment of our methods performance on predicting high-throughput phenotype data. As pointed out by the reviewer, analyzing high-throughput phenotype data could allow us to more quantitatively benchmark the ability of our method to predict specific metabolic auxotrophies (or inabilities to produce certain metabolites). We have identified a promising dataset that contains a large number of *E. coli* knockout mutants tested on different environments containing variable nutrients (Tohsato et al., 2010, J. Bioinform. Comput. Biol.). We have contacted the authors of this paper in the hope that they will share the raw data of their study with us, in which case we would pursue an analysis wherein we analyze this data to provide a more-high throughput and quantitative clarification of our PM calculations. This point is further elaborated in our specific response to reviewer #3 comment 1.

Part 2 – Point by point response to reviewers

Reviewer #1:[…]1) What is the difference between PM and% completion of biosynthetic pathway?

This is an important question that we propose to address through two different analyses. The first is a detailed analysis of a known (linear) biosynthetic pathway, biosynthesis of the amino acid L-histidine in *E. coli*. In a preliminary analysis (see proposed Figure 1 in Author response image 1 below) we analyzed this biosynthetic pathway by “knocking-out” different metabolic reactions and calculating our Producibility Metric (PM). One can see that the PM distinguishes between the effects of removing different reactions along the pathway based on their proximity to the target metabolite, L-histidine. Thus the PM metric is able to capture the impact of the specific location of the knockout and the topology of the metabolic pathway, thus carrying information about the chance that additionally available metabolites could recover the function of the broken pathway. In contrast, in several other possible approaches (including pathway completeness, as well as flux balance analysis, or network expansion) we expect the outcome of such a knock-out analysis to yield the same final output irrespective of the gene location, and thus provide a different and less informative prediction. For the full revision, we will show this analysis in more detail, and compare its outcome with the other methods mentioned above.

The second analysis that we propose goes beyond a simple linear pathway, and constitutes a more general assessment of the relationship between PM and biosynthetic pathway% completeness. For this analysis we will utilize our large set of oral microbiome organisms. For several metabolites with well described biosynthetic pathways, such as amino acids, we will determine the% completion across these strains and make a direct comparison with the PM value. We do expect to see some correlation between the% completion of a pathway and the PM value. However, as demonstrated with the above analysis, we do not expect them to be redundant.

We would also like to emphasize that we believe it is an advantage of our method that it does not rely on previously defined biosynthetic pathway annotations and instead seeks to use the entire metabolic network structure to define biosynthetic capabilities. While this prior knowledge can be quite useful in many contexts, specific biosynthetic routes can cross the boundaries of annotated pathways, making pathway completeness uninformative. A prior notable example of this effect is the discovery of an alternative pathway to bypass a TCA cycle gene impairment (Frezza et al.,2011). This pathway connects in an unexpected way two distinct pathways, generating a new crucially important and experimentally validated type of connectivity that would be missed from regular pathway-based analysis. Another such example, in our current data, is the case of the arginine deiminase pathway and the urea cycle, which contain several overlapping metabolites and reactions. In fact, we have noticed that KEGG mappings of TM7x metabolism often highlight the urea cycle as a result of TM7x containing the complete arginine deiminase pathway. In the revised manuscript we will describe this advantage of the Producibility Metric relative to pathway completeness, and point to the specific examples mentioned.

2) It's also not entirely clear to what extent their method improves upon the state of the art. In their Introduction, they say that two downsides that limit FBA (but not their method) are assumptions on environmental conditions and reconciling with stoichiometry-based constraints. I don't really understand their second limitation, but they sidestep the first limitation by assuming a constant P_in_ among all inputs. This seems similar to assuming constant concentration among all inputs for FBA, which is never shown.

We agree with the reviewer that the current description of how our method differs from FBA is not articulated clearly enough, and can be significantly improved. We have described in detail how we plan to address this issue in Part 1 Point B above, and will add further details to our manuscript to describe our method, and in particular the PM, *P_in_* and how they are related to FBA.

The first of the two “downsides” mentioned in the reviewer’s comment is related to a specific choice on how our approach can be employed for obtaining biological insight about an organism in the absence of specific information about the molecular composition of its environment. In a given FBA calculation, one necessarily needs to decide on a specific set of initial conditions, in the form of upper bounds to the fluxes associated with import of extracellular metabolites. Our method is instead statistical in nature, and computes the producibility of different metabolites as a function of the probability of availability of any given environmental metabolite (*P_in_*). Indeed, in the current version of the manuscript, we focused on a uniform *P_in_* across all metabolites, essentially mimicking a uniformly distributed ensemble of environments. However, in general, each metabolite *j* can have a different value *P_in_^(j)^*. An extreme case is the one of all *P_in_* = 1, which would be comparable to running FBA with all environmental metabolites available. In a revised version of the manuscript, we propose to show some specific examples to clarify how our method chooses statistical ensembles of input metabolites based on *P_in_* and how this choice differs from, but is related to, setting constant inputs for FBA. We will calculate the set of producible metabolites in the *E. coli* core model by using FBA under two different conditions, one where we include all input metabolites, as in a rich medium, and one where we include only input metabolites expected to be found in a glucose minimal medium. We note that these two scenarios correspond respectively to: (a) Rich Medium – setting *P_in_* values equal to 1 for all metabolites; (b) Minimal Glucose Medium – setting *P_in_* values equal to 1 for all metabolites found in glucose minimal medium and 0 for all others. We will then compare the metabolites that are producible in those two scenarios to the PM values obtained from varying *P_in_* between 0 and 1. We expect that metabolites that are producible in the minimal medium will have higher PM values than those that are producible only in the rich medium.

The second “downside” of other methods relative to our approach is related to the fact that our methods use of a full stoichiometric matrix to constrain results, as opposed to other approaches that are based on an adjacency matrix of metabolites. We realize that our explanation was not very clear, and we will make sure to clearly articulate this point in a revised version, which will include a figure supplement in which stoichiometry and adjacency are clearly compared to each other. In particular, in response to reviewer requests we will compare the predictions of our method with those of the NetCooperate method which utilizes an adjacency matrix (Levy et al.,2015; Levy and Borenstein, 2013). We will describe how our results differ from this method in terms of sensitivity and specificity. More specifically, we expect our method to be more specific in its predictions of exchanged metabolites and ability to discriminate different types of interactions than the NetCooperate method. Some results supporting this point are shown in preliminary Figure 3 (Author response image 3) below.

Reviewer #2:[…]1) The method assumes a uniform distribution of possible substrate uptake combinations – which is not necessarily biologically observed. There are hierarchies – e.g. preference of glucose over other C-sources in most organisms.

This is a very interesting point. We first would like to clarify that the assumption of uniform *P_in_* for our analysis was chosen specifically in order to assess environmental robustness across a uniformly sampled ensemble (see also Part 1, Point B above). Our method, however, can easily be implemented using variable *P_in_* values that represent any assumed prior distribution on input metabolites. We propose an analysis that will utilize variable *P_in_* values in order to further demonstrate the utility of our method for a specific biological example where variable assumptions about the environments may be of interest. We will analyze saccharolytic vs. proteolytic organisms from our set of oral microbiome organisms, which have key differing metabolic capabilities and implications in oral health and disease (Takahashi, 2015). Using variable *P_in_* values, we will define a sugar rich and an amino acid rich environment, and we will calculate the PM of all biomass components for both saccharolytic and proteolytic organisms in both. The sugar-rich environment will have sugars with fixed *P_in_* = 1 (i.e. they are always included in the environment) and the amino acid rich environment will have *P_in_* = 1 for amino acids. By fixing *P_in_* for these metabolites the PM will then become a measure of how many metabolites, beyond those that are fixed on, need to be added to the environment to produce a given target metabolite. These results will allow us to determine whether our method can capture the environmental preference and metabolic processes of these two classes of organisms from the human oral microbiome. This analysis is further described in prospective Figure 4, which has yet to be generated.

2) The finding that metabolic dependencies may exist in oral microbiome is not that surprising per se given that a large body of literature (including papers also from the Segre lab) already suggest that metabolic dependencies are likely widespread in microbial communities. Thus, it is not clear whether only the proposed method would be able to identify these.

Indeed, as expressed by the reviewer, a lot of literature exists about computational and empirical reports of inter-species dependencies in microbial communities. However, we believe that the reviewer will agree that what has been done uncovers with limited fidelity a minuscule fraction of the interactions likely present in nature. The challenge that remains is in characterizing and describing the mechanisms of these dependencies/interactions. We will add a statement emphasizing this point. Beyond this clarification, we will be happy to address more specifically the reviewer’s concern, i.e. show that our method can provide insight that is substantially different from the type of insight provided by other methods. We will compare our method to results obtained from three different alternative methods: (1) Pathway completeness, (2) Flux balance analysis, and (3) NetCooperate. These comparisons are, in part, addressed in the responses to other reviewer comments, and are described in detail in Part 1 point A, which outlines this major issue. We also note that, to our knowledge, ours is the first large-scale metabolic analysis of this type in the human oral microbiome. In this sense it provides valuable new insight into this microbial community and can further serve as a blueprint for analyzing other complex microbial communities.

3) Lack of experimental validation makes it difficult to assess whether the identified exchanges are happening in the system. At the least, a detailed analysis of systems with known metabolite exchanges (and how the proposed method there differs from the previous ones) should be provided.

This is a good point and we will attempt to address it as best we can. Unfortunately, there are limited systems where microbial metabolic dependencies and interactions have been unraveled experimentally. One example where there is some experimental knowledge that we will analyze is the exchange of organic acids in the oral microbiome. One such example is the exchange of lactic acid produced by *Streptococcus* and consumed by *Veillonella* species (Chalmers et al., 2008, J. Bact.). Another example is the exchange of succinate from *T. denticola* to *P. gingivalis* and isobutyric acid from *P. gingivalis* to *T. denticola* (Grenier, 1992, Infect. immun.). To analyze these examples, we will extend our method to not only calculate PM values for essential biomass components but also to include possible secreted metabolites such as organic acids. We will then utilize our method to calculate the PM for these different organisms and identify metabolic complements. Through this analysis we will confirm whether or not our method is able to capture these well-known microbial interactions. Furthermore, we will extend this analysis to calculate the PM of a set of possibly exchanged organic acids for all oral microbiome organisms in our catalog in order to gain broad insight into organic acid producibility and potentially predict new metabolic exchanges.

Another analysis we propose to perform is the study of a slightly more complicated microbial ecosystem, consisting of *E. coli* amino acid auxotrophs (Mee et al., 2014). In this work all pairs of *E. coli* single amino acid auxotrophs and all triplets of double auxotrophs where grown in co-culture to identify pairs and triplets that enabled syntrophic growth. We will re-analyze these *E. coli* auxotrophs and calculate their PM for all amino acids to determine if our method can predict the increased ability of growth-supporting auxotrophs to produce the amino acids that their syntrophic partners are lacking. We anticipate that certain amino acid auxotrophs, on top of being unable to produce a single specific amino acid, will have decreased robustness in their production of other amino acids that have similar biosynthetic pathways. This will lead to decreased syntrophic growth between auxotrophs that cannot robustly synthesize each other’s essential amino acids, and can be predicted by our PM calculations. For example, upon first inspection of the data from Mee et al. in Figure 1C we can see this pattern with regards to the L-methionine and L-cysteine auxotrophs (L-methionine and L-cysteine are the two sulfur containing amino acids and have similar biosynthetic pathways). While the L-methionine auxotroph has improved syntrophic growth with a number of other auxotrophs it has notably little syntrophic growth with the L-cysteine auxotroph. A further analysis of this data and comparison to our PM calculations will be presented in prospective Figure 5 (see legend below), to be included in the methods section of our manuscript as an additional validation of our method.

We will also analyze the performance of our metric in describing microbial co-occurrence patterns across the human oral microbiome. Although co-occurrence does not necessarily imply interactions, and mechanisms of interaction are not known for these cases, previous methods have been shown to be capable of identifying metabolic trends associated with microbial co-occurrence (Levy and Borenstein, 2013). We will use our method to analyze two different type of co-occurrence data and attempt to similarly identify metabolic trends. The first set of co-occurrence data is from sequencing data from the human microbiome project (Huttenhower et al., 2012, Nature). The human microbiome project included 9 different oral sites from which we will collect microbial co-occurrence data that can be compared to the oral microbiome strains that we have analyzed. The second form of co-occurrence data is from physical associations which have been observed through fluorescence microscopy (Welch et al., 2016). In this work the authors identified structure in supragingival plaque microbial communities from the oral microbiome. These structured communities suggest interactions that can similarly be compared with our PM metric to identify metabolic trends associated with microbial co-occurrence. The results of this analysis will be presented in prospective Figure 6 and included in a supplementary results section of our revised manuscript.

Reviewer #3:[…]Given the rate of generation of microbial genomes it would be very useful to have a better method to even approximately predict metabolic products of organisms. However, what the study lacks in my opinion, is quantitative evidence that the metric actually does translate into a prediction that the organism produces a given metabolite. They provide some anecdotal examples where it appears to work but without a more thorough analysis of what the metric means then it will only be useful in a qualitative sense i.e. a higher PM indicating that one organism is more likely to produce a metabolite than another. Metabolic modelling is admittedly not my field, but datasets capable of testing the metric, must exist. A possible example might be this collection of E coli strains and phenotypes defined in terms of growth conditions (Galardini et al., 2017, eLife). If such datasets are not available, then I feel the authors need to be more circumspect in their conclusions and concede that the metric may say something about the production of metabolites but without any certainty about how a given PM value translates into an actual probability of production.

We appreciate the reviewer’s comment, which made us realize that a number of subtle but important aspects of our analysis were not clearly explained in the manuscript. First of all, we recognize that we can do a better job at providing an intuition for the meaning of the PM metric, including a clarification that it is not meant to quantitatively capture rates of reactions (fluxes) (as done by FBA) but rather the fraction of environmental metabolites that an organism is expected to need in order to produce with high probability a given metabolite. Additional details of the intuition behind our definition are provided in Part 1, point B. We will further emphasize, in our revised manuscript, the nature of PM as a measure of robustness rather than a probability of production.

The most important concern expressed by the reviewer, however, seems to be the insufficient comparison with experimental data, which can lead one to legitimately wonder how accurate and valuable our predictions are. Irrespective of the outcome of the additional analyses we propose to perform for validating our predictions (see Part 1, point C above, and more details below), we will certainly take to heart the reviewer’s suggestion to more explicitly describe the possible limitations of our inferences. At the same time, this will also give us an opportunity to emphasize that, relative to existing methods, our approach provides a very unbiased estimate of the metabolic capabilities of a given organisms given its genome, avoiding some of the issues of gap-filling algorithms used for flux balance models.

In considering the reviewer’s recommendation of additional comparisons with experimental data, we have identified a few datasets whose analysis would enable a more substantial validation of our approach. These include a number of specific interactions between oral microbes, mediated by the exchange of organic acids, and a fairly rich dataset on amino acid exchange between *E. coli* auxotrophs. Both datasets are described in some more detail in response to reviewer #2 comment 3.

We also appreciate the reviewer’s suggestion of a high-throughput dataset that could be used to additionally assess our method, and we thank the reviewer for suggesting the interesting paper included. Unfortunately, the majority of the conditions tested in this paper are undefined or non-metabolic (mainly consisting of antibiotics or other stressors), and it is therefore not the best dataset for evaluating our method. In the spirit of the reviewer’s comments, we would like to suggest a similar high-throughput analysis using data from a different *E. coli* mutant phenotyping study that we identified (Tohsato et al., 2010, J. Bioinform. Comput. Biol.). This dataset includes phenotype data from 300 *E. coli* K12 mutants from the Keio collection grown on 1920 different conditions using Biolog Phenotypic Microarrays. Many of these conditions are metabolic in nature consisting of minimal media supplemented with different carbon, nitrogen, phosphorus, or sulfur sources. We have contacted the authors of this paper in the hope that they will share the raw data of their study with us, in which case we would pursue an analysis wherein we analyze this data to provide a more-high throughput clarification of PM and how exactly it relates to environmental robustness. In particular, we anticipate that *E. coli* knockouts with high PM for all essential biomass components will survive on a large number of variable nutrient environments, whereas knockouts with low PM for all essential biomass components will survive on a very small number of variable nutrient environments. Furthermore, analysis of the specific knockout/environment pairs could give us insight into the ability of our method to predict specific auxotrophies and metabolic requirements.

Other than the above caveat, I thought this was a good study but it was sparse on the practical details of the methodology. The GitHub repository consists of a selection of Matlab scripts without a clear description of how to apply them. What would be really useful is a complete walkthrough of the methodology from genome to PM values including for example ggkbase commands used to build the models. This is necessary if this work is to be properly reproducible. One final comment was that some evaluation of the impact of sample number on the calculated PM values would be a worthwhile addition. The authors mention that 50 samples of input metabolite combinations were used but no analysis of the impact of this choice on the PM values was given.

This is a good point and we appreciate the reviewer’s effort to help us make our method more reproducible. We will add to the GitHub repository example scripts that walk the readers through how to implement our algorithm and obtain our results. In addition to these example scripts, we will add a supplementary methods section that contains a procedural walk through on how to go from a genome to metabolic predictions using KBase (http://kbase.us) to reconstruct a model and our method to analyze it. Regarding the parameter choice for the algorithm, we note that those numbers were chosen by hand to improve algorithm performance. We will further describe the process of choosing these parameters and the impact on PM calculations including means and standard deviations of results.

PART 3 – Preliminary and prospective figures

**Author response image 1. respfig1:** Proposed Figure 1. Analysis of our Producibility Metric (PM) for individual gene deletions in the L-histidine biosynthetic pathway in *E. coli*. Note that the PM varies for all gene deletions tested, while other metrics such as pathway completeness, FBA, or network expansion would yield the same results for all knockouts. For pathway completeness the results would be 8 of 9 reactions present, and for FBA/network expansion Lhistidine would not be producible from an externally supplied metabolite fed into the top of the pathway. (**A**) The linear pathway was analyzed by knocking out individual reactions (squares) at different distances from L-histidine (red circle). (**B**) The PM was calculated for L-histidine and all other biomass components in the *E. coli* metabolic network, and it can be seen that the distance of the knock-out from L-histidine significantly affects its PM other biomass components are not affected. (**C**) The resulting PM from different knockouts (blue circles) was compared to the theoretical value for the PM of a metabolite with 1 to 9 individual precursors using our combinatorial theory (x’s) and was found to match. This result indicates that the additional co-factors utilized in this pathway (gray circles) have minimal impact on the final PM of Lhistidine.

These results will be further elaborated and included in a new figure in a supplementary methods section of our revised manuscript.

**Author response image 2. respfig2:** Proposed Figure 2. Comparison of the performance of our method relative to FBA in predicting metabolic properties of an organism (*E. coli*) as an increasing number of random gaps in the metabolic network are introduced..

The ability of the *E. coli* iJO1366 metabolic network to produce biomass under different conditions and constraints was analyzed for networks impacted by an increasing number of random perturbations that remove reactions in the network. Reactions were removed sequentially from the metabolic network. The resulting biosynthetic capabilities (on a rich medium) are plotted for the average of 10 different random runs, with error bars indicating standard error across runs. Flux balance analysis (FBA) calculations of the biomass flux become infeasible fairly quickly, whereas our metabolite producibility metric extends far beyond the FBA infeasibility region

An updated version of this figure will be included in a supplementary methods section of our revised manuscript.

**Author response image 3. respfig3:** Proposed Figure 3. Comparison of metabolic complementarity computed as an Advantage Metric based on our producibility metric (PM) vs. the NetCooperate complementarity score.

Complementarity metrics were calculated to summarize the potential for any one organism to provide metabolic products to another. The “advantage metric” was calculated using our PM metric that describes the metabolic advantage one organism can provide to another. This metric was plotted against an analogous complementarity metric calculated using NetCooperate for all directed pairs of 456 oral microbiome organisms. Log-scale histograms display the distribution of complementarities for each method. The histograms show that the advantage metric has a bi-modality that distinguishes between interactions involving fastidious/uncultivated organisms from those that do not, while the NetCooperate complementarity does not display a similar distribution.

The formula for the advantage metric is described below for organism A to B. The metric is calculated by summing over i = 1 to N metabolites.

A→B=maxPMiA,PMiB-PMiB∑i=1NPMiA

A subsequent version of this figure will be added to a supplementary results section of our revised manuscript as Figure 4—figure supplement 7.

Proposed Figure 4. To be generated. Analysis of variable *P_in_* values for saccharolytic and proteolytic oral microbiome organisms.

This figure will display the ability of our method to utilize variable *P_in_* values to provide biological insight into a set of organisms that may prefer a specific environment. An analysis of the PM values calculated for a set of saccharolytic and another set of proteolytic oral microbiome organisms in either a sugar or amino acid rich environment that has been altered by changing *P_in_* values will be presented.

This figure will be added to a supplementary results section of our revised manuscript.

Proposed Figure 5. To be generated. Analysis of *E. coli* amino acid auxotrophs.

This figure will display the ability of our method to predict metabolic interactions in a system with known metabolic dependencies. PM calculations for *E. coli* amino acid auxotrophs will be compared to data showing syntrophic growth among pairs and triplets of auxotrophs.

This figure will be added to a supplementary methods section of our revised manuscript as additional validation.

Proposed Figure 6. To be generated. Analysis of co-occurrence in the oral microbiome.

This figure will display the ability of our method to identify metabolic trends in microbial co-occurrence networks. Co-occurrence networks will be taken from two distinct sources. The first data source will be cooccurrence from sequencing data from the human microbiome project. The second will be physical associations from fluorescence microscopy data. We will further display how our metric, describing metabolic properties, can be used to describe metabolic trends in these co-occurrence networks.

This figure will be added to a supplementary results section of our revised manuscript.

[Editors' note: the authors’ plan for revisions was approved and the authors made a formal revised submission.]

Section 1 (Response to final recommendations)

The Reviewing Editor stated:"I think that I know how authors can fix their problems from a theoretical aspect. They can start with a complete model as the "ground truth" model, but hide or mis-annotate various% links to mimic our incomplete knowledge of the system, and compare their method with existing methods. This way, they will always have a "ground truth" curve for any metric of interest to biologists. Of course, the task of assigning missing links will need to be repeated many times and in double-blind fashion to gain statistical power. I suspect that different methods might show tradeoff. For example, in their Figure 1, it is not clear whether the extra info provided by PM is useful to a biologist if all mutants are auxotrophic. Any tradeoff between methods is also valuable to know. Authors can also use this ground truth model to illustrate problems of gap filling etc."

We thank the Reviewing Editor for this suggestion, which we addressed by analyzing a well curated metabolic network (“ground truth”) that has been degraded to different degrees (“miss-annotated”). This new analysis is presented in the main text, in a new section of the Results titled “Producibility analysis shows improved tolerance to missing reactions compared to flux balance analysis”. This section is accompanied by a newly added main figure (Figure 3). For this analysis, we used the well-curated *E. coli* iJO1366 metabolic network as our ground truth model, which we perturbed to different degrees by randomly removing reactions. This was repeated 50 times for 5 different levels of random reaction removal generating 250 different degraded metabolic networks. We analyzed these networks with FBA and with our producibility metric. This allowed us to show that the producibility metric is more tolerant of removed reactions in its predictions than FBA. Thus, this analysis illustrates that the producibility metric provides useful information on growth capabilities for draft metabolic networks that would need to be gap-filled in order to be analyzed with FBA.

Another reviewer said:"I agree this is a very comprehensive plan and they do seem to have taken our comments to heart and developed a proposal to address them. I was pleased with their suggestions regarding an alternate dataset of E. coli phenotypes to test their metric and the improved GitHub repository."

As described in Section 2, in response to reviewer #3’s comments 1 and 2, we have addressed in detail the comments made in the original review letter, expanding on our initial plan for a revision. This includes a number of new analyses including several that relate our method to experimental phenotypic data, both in *E. coli* (Results section “Metabolite producibility points to putative metabolic mechanisms for *E. coli* auxotroph co-cultures”) and in the human oral microbiome (Results section “Organic acid production predicted for human oral microbiome organisms”, and Results section “Metabolic similarity correlates with microbial co-occurrence in the human oral microbiome”). Additionally, we have improved the GitHub repository by adding a detailed example of how to run our code.

The last reviewer said:

"I agree that the proposal is comprehensive and I like the Reviewing Editor's simulation idea. This also reminds me that the authors' should comment on the ensembleFBA approach which does address the main criticisms of the gap-filling. Furthermore, I think that a clear focus on oral microbiome would benefit the study much more rather than claiming general validity that is very difficult to substantiate based on their proposed plan – after all, E. coli is one organism and by no means representative of natural diversity and community complexity. E. coli auxotroph complementation experiments are also unique in some sense as I haven't seen many other studies showing similar data in other organisms. So, overall, I think recommending to focus on oral microbiome and reducing the general claims would put the paper on more solid foundation I believe."

We appreciated this comment and recommendation from the reviewer. We gladly included reference to the ensembleFBA approach (Biggs et al., 2015), as well as to the newly published CarveMe method that is also capable of ensemble model reconstruction (Machado et al., 2018). We agree that these approaches constitute promising methods for addressing the gap-filling problem, and we are happy to mention them in the Introduction of the paper.

Regarding the more general comment on remaining focused on the oral microbiome and reducing our general claims, we greatly appreciate this advice. In balancing other reviewers’ comments with this suggestion, we have implemented some illustrative examples in *E. coli*, including an analysis of *E. coli* auxotroph co-cultures and the Reviewing Editor's simulation idea in *E. coli*; however, we have focused most other new analyses on oral microbiome organisms. New analyses focused on the oral microbiome include: (i) observing the relationship between the completeness of the histidine biosynthetic pathway and our metric (Results section “Producibility of metabolites differs from pathway completeness and captures minimal precursor set structure”), (ii) investigating organic acid production (Results section “Organic acid production predicted for human oral microbiome organisms”), (iii) implementing our method in an environment specific manner with the analysis of a saccharolytic and a proteolytic organism (Results section “Metabolite producibility in a protein vs. carbohydrate enriched environment”), and (iv) associating our metric with co-occurrence from 16S rRNA sequencing data from the human oral microbiome (Results section “Metabolic similarity correlates with microbial co-occurrence in the human oral microbiome”). In all of these analyses we have sought to relate our findings with phenotypes that are relevant for the human oral microbiome whenever possible, trying to candidly illustrate what works well and what doesn’t. Additionally, we feel that the revised manuscript provides a better explanation for why and how our approach is particularly useful for addressing microbial community questions in cases where the specific metabolic composition of the environment is highly uncertain, such as in the study of natural human-associated microbiomes. This clarifies our motivation for focusing on the human oral microbiome in this study and provides insight into which scenarios our method is best suited for in the future.

Section 2 (Response to initial reviewer comments)

Reviewer #1:[…]1) What is the difference between PM and% completion of biosynthetic pathway?

This is an important question, which we have addressed through new analyses added to the Results section “Producibility of metabolites differs from pathway completeness and captures minimal precursor set structure” and Figure 2—figure supplement 2. We have additionally introduced a new paragraph in the Discussion addressing the difference between our method and alternative methods including biosynthetic pathway-based analysis. The specific changes are described in detail below:

The first analysis we performed was focused on looking in detail at the biosynthetic pathway for the amino acid histidine in *E. coli*. We focused on this pathway because it is overall linear, and thus more easily interpretable. We used the well curated *E. coli* iJO1366 metabolic network and we examined this biosynthetic pathway by “knocking-out” different metabolic reactions along the pathway and calculating our Producibility Metric (PM). We observed that the PM was able to capture the impact of the specific location of the knockout and the topology of the metabolic pathway. In contrast, looking at the% completion of the biosynthetic pathway would not provide this information.

A second analysis extended this concept to all 456 oral microbiome metabolic networks. Specifically, we calculated both the% completion of the histidine biosynthetic pathway (total number of reactions present) and the length (distance from histidine before a reaction is missing). We found that both of these metrics are correlated with the PM for histidine. However, the length of the intact portion of the biosynthetic pathway is more strongly correlated with the PM than with the% completion, corroborating, across a large set of metabolically diverse organisms, the result obtained for *E. coli*.

A deeper, and theoretically motivated portion was additionally introduced in the end of this newly added Results section, to further clarify how the PM differs from% completion of a biosynthetic pathway. We found that the PM for histidine of different knock-outs matched closely with our theoretical predictions based on combinatorics. In this new section, we point out that the combinatorial equations establish an intimate connection between the PM and the minimal precursor set structure of a particular target metabolite, and thus captures a complex measure of the multiplicity of biosynthetic routes through which the target metabolite can be produced. This theoretical analysis points out a profound meaning of the PM, which could be further explored in future work.

Finally, we added a section to the Discussion, to emphasize that our method has another important advantage over% completion: it does not rely on previously defined biosynthetic pathway annotations. While this prior knowledge can be quite useful in many contexts, specific biosynthetic routes can cross the boundaries of annotated pathways, making pathway completeness uninformative. We have highlighted a few scenarios from the literature and in our own analysis where this is the case, and where less biased network based approaches have an advantage.

2) It's also not entirely clear to what extent their method improves upon the state of the art. In their Introduction, they say that two downsides that limit FBA (but not their method) are assumptions on environmental conditions and reconciling with stoichiometry-based constraints. I don't really understand their second limitation, but they sidestep the first limitation by assuming a constant P_in_ among all inputs. This seems similar to assuming constant concentration among all inputs for FBA, which is never shown.

We agree with the reviewer’s comment that our original version of the manuscript did not clearly articulate how our method improves upon the state of the art. In the revised manuscript we have made changes to the Introduction and Discussion that clarify the novelty of our method, and we have added a number of analyses to the Results section where our method is directly compared with alternative methods.

The main novelty of our method, i.e. the statistical nature of the percolation approach, is now reflected in a modified Title and Abstract, and better explained in a revised, more straightforward version of Figure 1 (the original Figure 1 is still available as Figure 1—figure supplement 1). Furthermore, we updated Figure 2B to more clearly demonstrate the features of the PM; specifically, that it is independent from node degree and that recycled co-factors have minimal impact on PM. To better explain in general terms how our method improves upon the state of the art, we significantly altered the text of our Introduction. In this revised Introduction, we revisit the first limitation mentioned by the reviewer, by clarifying the rationale of assumptions on environmental conditions, i.e. the ability of our method to provide predictions in scenarios where the chemical environment of the microbes is unknown. We also note that our choice of constant *P_in_* was chosen as the most naïve choice by which we could statistically sample the environment. This approach is different than providing all nutrients with uniform uptake rate in regular FBA, because it samples an ensemble of environments, rather than assuming a single uniformly rich one. Additionally, for reviewer #2 comment 1, we have implemented an analysis where we use non-uniform *P_in_* to demonstrate the capacity of our method to incorporate environmental assumptions (Results section “Metabolite producibility in a protein vs. carbohydrate enriched environment”). Regarding the second limitation mentioned by the reviewer, about stoichiometry-based constraints, we discuss this in more detail in a modified portion of the Discussion, where we highlight the difference between networks defined by reaction stoichiometry vs. connectivity.

The reviewer’s concern about improving upon the state of the art was further addressed in detail by adding to the Results section several new analyses in which we compare our method with alternative ones. In addition to the comparison of our method with percent pathway completion, discussed in detail in the previous comment, we implemented the following new comparative analyses:

i) As described also in Summary Comment 1, a comparison with FBA, including an analysis of how FBA and the PM cope with a gradually increasing number of perturbations of a genome-scale metabolic network (mimicking the situation of non-gap-filled draft reconstructions), was addressed in the new section “Producibility analysis shows improved tolerance to missing reactions compared to flux balance analysis” of the Results and Figure 3.

ii) We also compared our method to several different metabolic network analysis methods in the context of microbial co-occurrence data in the new Results section “Metabolic similarity correlates with microbial co-occurrence in the human oral microbiome” and the newly added Figure 4—figure supplement 6. We showed that an inter-organism distance based on our PM score explains co-occurrences better than scores based on alternative methods we tested (as shown in the newly added Supplementary file 4). We further show that our approach more precisely separates fastidious/uncultivated microbes than an alternative approach in Figure 4—figure supplement 7. Further comments on how our method differs from and improves upon other potential approaches, as well as an extended discussion of limitations, have been added to two paragraphs in the Discussion.

Reviewer #2:[…]1) The method assumes a uniform distribution of possible substrate uptake combinations – which is not necessarily biologically observed. There are hierarchies – e.g. preference of glucose over other C-sources in most organisms.

We thank the reviewer for raising this point, which gave us the opportunity to better clarify two important aspects of our analysis. First, the assumption of uniform *P_in_* for our analysis was chosen specifically in order to assess biosynthetic capabilities across a uniformly sampled ensemble of environments, i.e. under the assumption of minimal knowledge about the chemical composition of the growth medium. We have emphasized this point in the revised Abstract, mentioned it in the Introduction, and clarified it in the revised “Analysis Method” section and the newly added Figure 1.

Second, while in the previous version of our manuscript we indeed only focused on the case of a uniform *P_in_*, we were glad to clarify and exemplify in the revised manuscript the feasibility of arbitrarily chosen, metabolite specific set of *P_in_* values, which encode assumptions about the environmental composition. In particular, to demonstrate this capability, we have recalculated the producibility metric (PM) for one saccharolytic and one proteolytic organism in a protein enriched environment and carbohydrate enriched environment. While we found only modest increases in the PM in both of the enriched environments, we did find that the producibility metric was further increased for the proteolytic organism in the protein enriched environment than it was for the saccharolytic organism. This analysis demonstrates how we can tailor our method to use variable *P_in_* values to ask interesting questions about an organism’s environmental dependencies. The analysis is presented as newly added Results section “Metabolite producibility in a protein vs. carbohydrate enriched environment” and Figure 4—figure supplement 5.

2) The finding that metabolic dependencies may exist in oral microbiome is not that surprising per se given that a large body of literature (incl. papers also from the Segre lab) already suggest that metabolic dependencies are likely widespread in microbial communities. Thus, it is not clear whether only the proposed method would be able to identify these.

Indeed, as expressed by the reviewer, the novelty of our work is not so much in pointing out the general possibility of inter-species metabolic exchanges, but rather in (i) the capacity of our method to produce predictions in a way that is specific to the underlying genome and molecular products, but at the same time robust to uncertainty in environmental composition and to missing gene annotations; and (ii) the production of a detailed atlas of biosynthetic capabilities and complementarities for a specific ecosystem of high interest for multiple health-related studies, namely the human oral microbiome. To our knowledge, ours is the first large-scale metabolic analysis of this type in the human oral microbiome. In this sense, it provides valuable new insight into this microbial community and can further serve as a blueprint for analyzing other complex microbial communities.

In the revised manuscript, however, in line with the reviewer’s comment, we made sure to better present our results in the context of previous analyses. In particular, in addition to more clearly stating the objectives of our work in the Introduction, we reference and describe in more detail prior literature on computational and empirical analyses of inter-species dependencies in microbial communities. Beyond this clarification, we have also added a number of analyses that compare our method to different alternative methods including: pathway completeness (Results section “Producibility of metabolites differs from pathway completeness and captures minimal precursor set structure”, Figure 2—figure supplement 2), flux balance analysis (Results section “Producibility analysis shows improved tolerance to missing reactions compared to flux balance analysis”, Figure 3), and alternative metabolic network analysis methods, such as NetCooperate used in a similar analysis (Levy and Borenstein, 2013), through an analysis of microbial co-occurrence data from the human oral microbiome (Results section “Metabolic similarity correlates with microbial co-occurrence in the human oral microbiome”, Supplementary file 4, Figure 4—figure supplement 6, Figure 4—figure supplement 7).

3) Lack of experimental validation makes it difficult to assess whether the identified exchanges are happening in the system. At the least, a detailed analysis of systems with known metabolite exchanges (and how the proposed method there differs from the previous ones) should be provided.

This is an excellent point. We are grateful to the reviewer for the suggestion, which prompted us to perform a number of analyses to assess the capacity of our method to relate to experimental measurements, and to compare our approach with previously published ones. We believe that this process ended up strengthening our manuscript significantly.

First, we examined an experimental dataset on microbial co-culture growth. This new analysis is described in a new section of the Results titled “Metabolite producibility points to putative metabolic mechanisms for E. coli auxotroph co-cultures”, Figure 3—figure supplement 1. For this new analysis, we used *E. coli* auxotroph co-culture data from Wintermute and Silver (2010). This dataset has the advantage of using genetically well-defined *E. coli* strains with specific known auxotrophies, such that corresponding genome scale models can be appropriately constructed, and metabolic exchanges can be inferred with reasonable confidence. We analyzed this system in detail to provide insight into the predictive capacity of our method. The comparison of model predictions with experimental data led to a number of observations, reported in the manuscript, and summarized briefly as follows: (i) we found that for several auxotrophs in the tryptophan pathway, the PM value for tryptophan was correlated with the auxotrophs’ ability to be supplemented by other auxotrophs; (ii) we observed that metabolically similar organisms (as measured by their PM difference across all biomass components) tend not to supplement each other’s growth; (iii) in the new section we also point out some exceptions to the trends mentioned above, including increased synergistic growth for at least one set of auxotrophs with knockouts in the same biosynthetic pathway, which highlights the complexity of this experimental data and the limitations of our method.

In a second newly added analysis, we utilized our metric, as well as previously published metrics, to describe microbial co-occurrence patterns across the human oral microbiome. The results of these analyses are presented in a new Results section titled “Metabolic similarity correlates with microbial co-occurrence in the human oral microbiome”, accompanied by a newly added Supplementary file 4. Although co-occurrence does not necessarily imply interactions, and mechanisms of interaction are generally unknown, previous methods have been shown to be capable of identifying metabolic trends associated with microbial co-occurrence in the human microbiome (Levy and Borenstein, 2013) We used our method to analyze co-occurrence from metagenomics data from 7 different oral sites in the human microbiome project. Co-occurrence data calculated using both the SparCC method and Pearson’s correlation were taken from Friedman and Alm (2012). We compared our method with a number of different metabolic network analysis methods, including NetSeed, NetCooperate, and NetCmpt used in Levy and Borenstein, as well as simple metrics based on metabolic reaction distance and Jaccard’s distance. We found that our metabolic network metric, based on the difference in PM between two organisms for all biomass components, was the most consistently anticorrelated with co-occurrence data (irrespective of co-occurrence estimate method), showing that similar organisms tend to co-occur. This result corroborates the habitat filtering hypothesis previously proposed by Levy and Borenstein. We also found that our method, and other methods, correlated more consistently with co-occurrence inferred from the SparCC method than those inferred from Pearson’s correlation, which further corroborates the claims of Friedman and Alm that Pearson’s correlations can fail to capture true co-occurrence in compositional data. Additionally, we found that our PM based metric maintained correlation with co-occurrence when controlling for other metrics through partial correlations while the inverse was not always true. This analysis highlights our metric’s ability to capture metabolic similarities between organisms, that correlate with co-occurrence, beyond what is captured by existing metabolic network analysis methods.

Finally, we extended our calculation of PM to include organic acid production. This analysis is presented in a new Results section titled “Organic acid production predicted for human oral microbiome organisms”, Figure 4—figure supplement 4. Through this analysis, we used our approach to determine whether it could predict organic acid production in human oral microbiome organisms. In particular, we were able to identify trends in organic acid production by different genera of oral microbiome organisms. We found that the *Fusobacteria* genera had increased PM for butyrate, a result with important implications in oral health, and which is supported by independent transcriptomic data from the literature (Jorth et al., 2014). Additionally, we found that some species of the *Porphyromonas* and *Prevotella* genera, which have been implicated in periodontal disease due to production of organic acids (Takahashi, 2015), had increased PM for butyrate as well. These results, while not directly proving exchange of a given metabolite across a specific organism pair, provide valuable information about putative secretions that could mediate these interactions.Reviewer #3:[…]Given the rate of generation of microbial genomes it would be very useful to have a better method to even approximately predict metabolic products of organisms. However, what the study lacks in my opinion, is quantitative evidence that the metric actually does translate into a prediction that the organism produces a given metabolite. They provide some anecdotal examples where it appears to work but without a more thorough analysis of what the metric means then it will only be useful in a qualitative sense i.e. a higher PM indicating that one organism is more likely to produce a metabolite than another. Metabolic modelling is admittedly not my field, but datasets capable of testing the metric, must exist. A possible example might be this collection of E. coli strains and phenotypes defined in terms of growth conditions (Galardini et al., 2017, eLife). If such datasets are not available then I feel the authors need to be more circumspect in their conclusions and concede that the metric may say something about the production of metabolites but without any certainty about how a given PM value translates into an actual probability of production.

We appreciate the reviewer’s comment, which raises a very good point. The most important concern expressed by the reviewer seems to be the insufficient comparison with experimental data, which can lead one to legitimately wonder how accurate and valuable our predictions are. Throughout our response to all reviewer comments, we have taken to heart this concern, by implementing a number of new analyses that compare our predictions with different kinds of experimental data, as well as with previously published approaches. In presenting these new analyses, we have shown the strengths of our approach, but we have also explicitly described the possible limitations of our inferences. We believe that these additional analyses will provide the readers with a clearer perspective of the type of microbial ecology questions for which our method can provide valuable insight.

The first analysis we performed to illustrate the value of our method was a comparison with the simpler approach of quantifying the percent completion of a pathway as a measure of biosynthetic capability. This analysis was also described in further detail for reviewer #1 comment 1, and is presented in the Results section “Producibility of metabolites differs from pathway completeness and captures minimal precursor set structure”, Figure 2—figure supplement 2. For this analysis, we analyzed the histidine biosynthetic pathway in *E. coli* in detail, calculating both the Producibility Metric (PM) and the% completion of the biosynthetic pathway for different knockouts along the linear pathway. Through this analysis we were able to illustrate the difference between the PM and a simpler pathway based analysis, giving additional insight into the meaning of the PM.

Next, we compared our method with flux balance analysis (FBA) for a genome-scale metabolic network with varying degrees of perturbations generated by randomly removing reactions from the network. This analysis is described in further detail in the Reviewing Editor’s summary comment and is presented in the Results section “Producibility analysis shows improved tolerance to missing reactions compared to flux balance analysis” and newly added main text Figure 3. Through this analysis we demonstrated that the PM is more robust to missing reactions than traditional FBA and can therefore provide additional insight into draft metabolic networks with gaps. Again, this analysis provided additional context to the meaning of the PM and its applicability for analyzing metabolic networks.

Next, we introduced several new analyses that compare our PM predictions to experimental data. These analyses provide insight into how the PM can be used to analyze and interpret biological data. The first of these was an analysis of *E. coli* auxotroph co-cultures presented in Results section “Metabolite producibility points to putative metabolic mechanisms for *E. coli* auxotroph co-cultures”, Figure 3—figure supplement 1. In this section, we use the PM to analyze co-culture synergistic growth data of different *E. coli* auxotrophs from Wintermute and Silver (2010). The comparison of PM predictions with experimental data led to a number of observations, reported in the manuscript, and summarized briefly as follows: (i) we found that for several auxotrophs in the tryptophan pathway, the PM value for tryptophan was correlated with the auxotrophs’ ability to be supplemented by other auxotrophs; (ii) we observed that metabolically similar organisms (as measured by their PM difference across all biomass components) tend not to supplement each other’s growth; (iii) in the new section we also point out some exceptions to the trends mentioned above, including increased synergistic growth for at least one set of auxotrophs with knockouts in the same biosynthetic pathway, which highlights the complexity of this experimental data and the limitations of our method.

We next extended our analysis of the human oral microbiome to predict organic acid production. This analysis is presented in Results section “Organic acid production predicted for human oral microbiome organisms”, Figure 4—figure supplement 4. We were able to identify trends in organic acid production by different genera of oral microbiome organisms. We found that the *Fusobacteria* genera had increased PM for butyrate, a result with important implications in oral health, and which is supported by independent transcriptomic data (Jorth et al., 2014). Additionally, we found that some species of the *Porphyromonas* and *Prevotella* genera, which have been implicated in periodontal disease due to production of organic acids (Takahashi, 2015), had increased PM for butyrate as well.

Finally, we added a large analysis of microbial co-occurrence data from metagenomic sequencing data. This analysis is presented in the new Results section “Metabolic similarity correlates with microbial co-occurrence in the human oral microbiome”, Supplementary file 4. In this analysis, we analyzed co-occurrence data from Friedman and Alm (2012) and calculated the correlation of co-occurrence with various inter-organism metabolic metrics calculated using our PM and alternative methods. We showed that an inter-organism distance based on our PM score explains co-occurrences better than scores based on any alternative method that we tested. Furthermore, there was a strong negative correlation between PM based inter-organism distance and co-occurrence indicating that organisms with similar biosynthetic capabilities tend to co-occur. Our finding corroborates and strengthens previous similar analyses of co-occurrence data from the human microbiome (Levy and Borenstein, 2013), and demonstrates the ability of our method to provide insight into large-scale top down data such as microbial co-occurrence from metagenomic sequencing.

We also appreciate the reviewer’s suggestion of a paper (and corresponding high-throughput dataset) that could be in principle used to additionally assess our method. Unfortunately, the majority of the conditions tested in this paper are undefined or non-metabolic (mainly consisting of antibiotics or other stressors), and it is therefore not the best dataset for evaluating our method. The alternative option we had considered, using a different dataset of similar type (from Tohsato et al., 2010, J. Bioinform. Comput. Biol.), ended up not being fruitful either, as we were unable to obtain this dataset. While this would have been a nice addition, we hope that the large number of new analyses we ended up performing to address this and other concerns will be deemed satisfactory.

Other than the above caveat, I thought this was a good study but it was sparse on the practical details of the methodology. The GitHub repository consists of a selection of Matlab scripts without a clear description of how to apply them. What would be really useful is a complete walkthrough of the methodology from genome to PM values including for example ggkbase commands used to build the models. This is necessary if this work is to be properly reproducible. One final comment was that some evaluation of the impact of sample number on the calculated PM values would be a worthwhile addition. The authors mention that 50 samples of input metabolite combinations were used but no analysis of the impact of this choice on the PM values was given.

This is a good point and we appreciate the reviewer’s effort to help us make our method more reproducible. We have added an example script to the GitHub page that can be run to calculate the producibility metric (PM) for a small subset of metabolic networks and metabolites. We also include code that could be used to calculate the PM for the 88 biomass metabolites and 456 oral microbiome metabolic networks analyzed throughout the paper. However, we note that this would take a long time and is best performed in parallel on a shared computing cluster, as described in the Materials and methods section.

Regarding the parameter choices, we note that these parameters were chosen by hand to provide adequate performance for our algorithm, which we have further emphasized in the Materials and methods section of our manuscript.